# A Practical Guide to Sample-based Statistical Distances for Evaluating Generative Models in Science

**Sebastian Bischoff**[* 1,2,3,4], **Alana Darcher**[5], **Michael Deistler**[1,2], **Richard Gao**[1,2], **Franziska Gerken**[6], **Manuel Gloeckler** [*†1,2], **Lisa Haxel**[1,2,3,7], **Jaivardhan Kapoor** [*1,2], **Janne K Lappalainen**[1,2], **Jakob H Macke**[1,2], **Guy Moss**[1,2], **Matthijs Pals**[* †1,2], **Felix Pei**[1,2], **Rachel Rapp**[1,2], **A Erdem Sağtekin**[1,2], **Cornelius Schröder**[1,2], **Auguste Schulz**[*1,2], **Zinovia Stefanidi**[1,2], **Shoji Toyota**[8], **Linda Ulmer**[1,2], and **Julius Vetter**[1,2]

[1]Machine Learning in Science, Excellence Cluster Machine Learning, University of Tübingen, Tübingen, Germany
[2]Tübingen AI Center, University of Tübingen, Tübingen, Germany
[3]University Hospital Tübingen, University of Tübingen, Tübingen, Germany
[4]M3 Research Center, University Hospital Tübingen, Tübingen, Germany
[5]University Medical Center of Bonn, Bonn, Germany
[6]Dynamic Vision and Learning Group, Technical University of Munich, Munich, Germany
[7]Hertie Institute for Clinical Brain Research, Tübingen, Germany
[8]The Institute of Statistical Mathematics, Tokyo, Japan[‡]

**Reviewed on OpenReview:** https://openreview.net/forum?id=isEFziui9p

## Abstract

Generative models are invaluable in many fields of science because of their ability to capture high-dimensional and complicated distributions, such as photo-realistic images, protein structures, and connectomes. How do we evaluate the samples these models generate? This work aims to provide an accessible entry point to understanding popular sample-based statistical distances, requiring only foundational knowledge in mathematics and statistics. We focus on four commonly used notions of statistical distances representing different methodologies: Using low-dimensional projections (Sliced-Wasserstein; SW), obtaining a distance using classifiers (Classifier Two-Sample Tests; C2ST), using embeddings through kernels (Maximum Mean Discrepancy; MMD), or neural networks (Fréchet Inception Distance; FID). We highlight the intuition behind each distance and explain their merits, scalability, complexity, and pitfalls. To demonstrate how these distances are used in practice, we evaluate generative models from different scientific domains, namely a model of decision-making and a model generating medical images. We showcase that distinct distances can give different results on similar data. Through this guide, we aim to help researchers to use, interpret, and evaluate statistical distances for generative models in science.

## 1 Introduction

Generative models that produce samples of complex, high-dimensional data, have recently come to the forefront of public awareness due to their utility in a variety of scientific, clinical, engineering, and commercial domains (Bond-Taylor et al., 2021). Prominent examples include StableDiffusion (SD) and DALL-E for generating photo-realistic images (Rombach et al., 2022a), WaveNet (Oord et al., 2016) for audio synthesis,

---

[*]Co-organizer
[†]Corresponding authors: manuel.gloeckler@uni-tuebingen.de, matthijs.pals@uni-tuebingen.de
[‡]Current affiliation: Department of Advanced Information Technology, Kyushu University, Fukuoka, Japan
  Authors listed in alphabetical order.

and Generative Pre-trained Transformer (GPT; Radford et al. 2018; 2019; Brown et al. 2020) for text generation. In addition to the recent surge in generative models, many scientific disciplines have a long history of developing data-generating models that capture specific processes. In neuroscience, for example, the occurrence of action potentials is commonly modeled at varying levels of detail (e.g., single neuron voltage dynamics; Hodgkin & Huxley 1952, or at a phenomenological level; Pillow et al. 2008), whereas in, e.g., astrophysics there exist various models to simulate galaxy formation (Somerville & Davé, 2015). Along with generating novel synthetic samples, generative models can be leveraged for specific tasks, such as sample generation conditioned on class labels (e.g., diseased vs. healthy brain scans Pombo et al. 2021, molecules that can or cannot be synthesized; Urbina et al. 2022, class-conditional image generation; van den Oord et al. 2016; Dockhorn et al. 2022), forecasting future states of a dynamical system (e.g., Jacobs et al. 2023; Brenner et al. 2022; Pals et al. 2024), generating neural population activity conditioned on visual stimuli or behavior (e.g., Molano-Mazon et al. 2018; Bashiri et al. 2021; Schulz et al. 2024; Kapoor et al. 2024), data imputation (e.g., Vetter et al. 2023; Lugmayr et al. 2022), data augmentation for downstream tasks (Rommel et al., 2022), and many more (see also Table S1).

These powerful capabilities are enabled by the premise that generative models can produce samples from the high-dimensional distribution from which we assume our dataset was sampled. The dimensions can correspond to anything from individual pixels or graphs to arbitrary features of physical or abstract objects. When aiming to build generative models that better capture the true underlying data distribution, we need to answer a key question: *How accurately do samples from our generative model mimic those from the true data distribution?*

Manual inspection of generated samples can be a good first check, e.g., in image or audio generation, where we can directly assess the visual likeness or sound quality of the samples (Gerhard et al., 2013; Vallez et al., 2022; Jayasumana et al., 2023). In general, however, we would like to quantitatively compare the similarity of distributions, for instance to benchmark different generative models. Many measures have been proposed that assess the similarity of two distributions based on various aspects of their moments or the overall probability density (Fig. 1). Some of these measures require likelihood evaluations, as is possible with generative models such as Gaussian Mixture Models, Normalizing Flows, Variational Autoencoders, autoregressive models or diffusion models (Bishop, 2006; Papamakarios et al., 2021; Box et al., 2015; Kingma & Welling, 2014; Yenduri et al., 2024; Ho et al., 2020; Song et al., 2021). However, many contemporary machine learning models (e.g., Generative Adversarial Networks and Energy-Based Models; Goodfellow et al. 2014; Rezende et al. 2014; Hinton et al. 2006) and scientific simulators (e.g., single neuron voltage dynamics; Hodgkin & Huxley 1952) only define the likelihood *implicitly*, i.e., we can not explicitly evaluate their likelihood. Statistical distances that can be computed based on samples only are therefore invaluable for comparing both classes of generative models (and to real data) in scientific contexts.

Here, we present a guide that aims to serve as an accessible entry point to understanding commonly used sample-based statistical distances. Towards this goal, we provide explanations, comparisons, and example applications of four commonly applied distances: Sliced-Wasserstein (SW), Classifier Two-Sample Tests (C2ST), Maximum Mean Discrepancy (MMD), and Fréchet Inception Distance (FID). With these resources, we aim to empower researchers to choose, implement, and evaluate the usage and outcomes of statistical distances for generative models in science. Note that, with *distance*, we do not necessarily refer to a distance metric in the mathematical sense (i.e., satisfying symmetry and the triangle inequality) but to a general measure of dissimilarity between two distributions (however, some considered distances are in fact metrics).

The general and didactic nature of this guide means it can neither be comprehensive nor provide a clear-cut answer as to which distance is 'best', as there are numerous choices, and the 'ideal' one strongly depends on the domain of application. Previous articles have provided useful and extensive reviews for specific use cases. For example, Theis et al. (2016) discusses commonly used criteria for evaluating generative models of images, Borji (2019); Xu et al. (2018); Yang et al. (2023) compare metrics specific to evaluating GANs (including, e.g., specialized variants of the FID), Basseville (2013) provides an extensive overview of previous works on divergences, and Gibbs & Su (2002) analyze the theoretical relationships among 'classic' distances (including, e.g., Wasserstein and KL-divergence). Rather, by going through four different classes of sample-based distances in detail and systematically comparing them on synthetic and real-world applications,

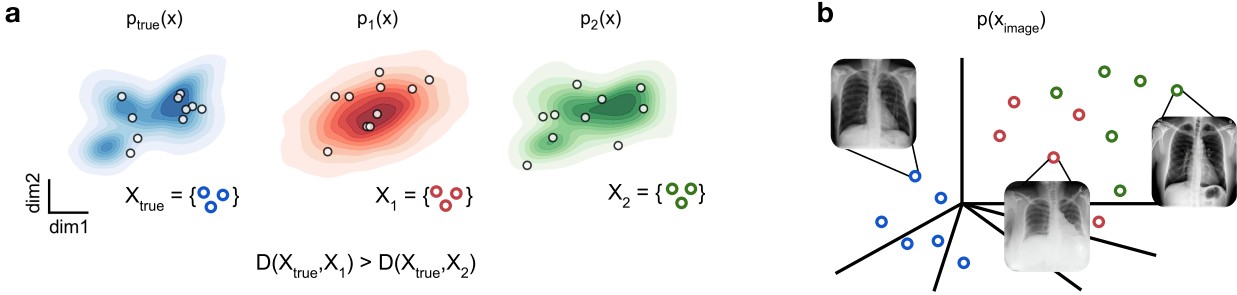

Figure 1: **The need for statistical distances in scientific generative modeling. (a)** An example target distribution, $p_{true}(x)$, and two learned distributions ($p_1(x)$ and $p_2(x)$) of different models trained to capture $p_{true}(x)$. All three distributions share the same mean and marginal variances, despite having distinct shapes. However, an appropriate sample-based distribution distance $D$ can determine that $p_2(x)$ is more similar to $p_{true}(x)$. **(b)** Scientific applications often require evaluating high-dimensional distributions, such as distributions of images or tabular data. In this example, each point represents an X-ray image, where each dimension is one pixel.

we aim to provide a solid foundation for navigating the extensive literature on statistical distances, and to enable readers to reason about other related distances not covered here.

The outline of this guide is as follows: First, we provide an intuitive and graphical explanation for each of the four distances (Section 2). We then perform a systematic evaluation of their robustness as a function of dataset size, data dimensionality, and other factors, such as data multimodality (Section 3). Finally, we demonstrate how these distances can be applied to compare generative models in different scientific domains (Section 4): We evaluate low dimensional models of decision making in behavioral neuroscience and generative models of medical X-ray images. We show the importance of using multiple complementary distances, as distinct distances can give different results when comparing the same sets of samples.

## 2 Sample-based statistical distances

In this section, we provide an overview of four classes of sample-based statistical distances commonly used in machine learning literature. Each class takes a different approach to overcoming the challenges inherent in comparing samples from high-dimensional and complex distributions. Throughout the section, we assume that we want to evaluate the distance between two datasets of samples, denoted as $\{x_1, x_2, \ldots, x_n\} \sim p_1(x)$ and $\{y_1, y_2, \ldots, y_m\} \sim p_2(y)$, where $p_1(x)$ and $p_2(y)$ are two probability distributions. These can be either two generative models, or a generative model and the underlying distribution of the observed data.

### 2.1 Slicing-based: Sliced-Wasserstein (SW) distance

Computing distance between distributions suffers from the *curse of dimensionality*, where the computational cost of computing the distance increases very rapidly as the dimensionality of the data increases. This problem is especially restricting when the distance is used as part of a loss function in optimization problems, since in this case it must be evaluated many times. This has prompted the notion of "sliced" distances, which have become increasingly popular in recent years (Kolouri et al., 2019; Nadjahi et al., 2020; Goldfeld & Greenewald, 2021). The main idea behind slicing is that for many existing statistical distances we can efficiently evaluate the distance in low-dimensional spaces, especially in a single dimension. Therefore, the "slices" are typically one-dimensional lines through the data space (Fig. 2a). All data points from each distribution are projected onto this line by finding their nearest point on the line, giving a one-dimensional distribution of projected data points (Fig. 2b). The distance measure of interest between the resulting one-dimensional distributions can then be computed efficiently. However, computing the random projection could lead to an unreliable measure of distance as distinct distributions can produce the same one-dimensional

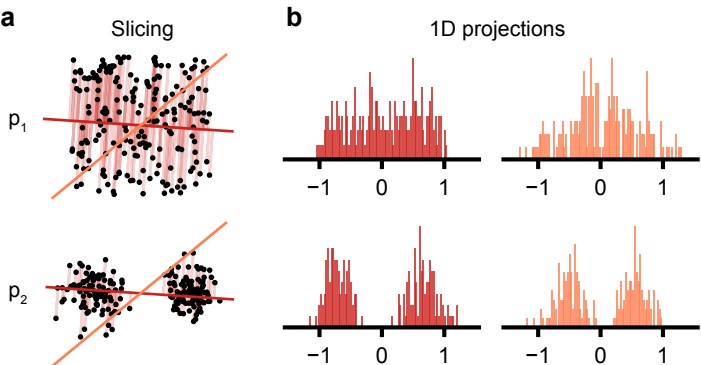

Figure 2: **Schematic for the Sliced-Wasserstein distance.** **(a)** Samples from two two-dimensional distributions along with example slices. The "slicing" is done by sampling random directions from the unit sphere and projecting the samples from the higher-dimensional distribution onto that direction. **(b)** One-dimensional projections of the two distributions corresponding to the two random slices in (a). For each pair of projections, the empirical Wasserstein distance is computed. Unlike in higher dimensions, this can be done efficiently for one-dimensional distributions.

projections. Therefore, we repeat the slicing process for many different slices and average the resulting distances. More formally, we compute the expected distance in one dimension between the projections of the respective distributions onto (uniformly) random directions on the unit sphere. As long as the distance of choice is a valid metric in one dimension, the sliced distance defined in this way is guaranteed to be a valid metric as well (Nadjahi et al., 2020, Proposition 1 (iii)). The most popular example of a sliced distance metric is the Sliced-Wasserstein (SW) distance (Fig. 2). The Wasserstein distance and its sliced variant have several attractive properties: They can be computed differentiably; their computations do not rely heavily on choices of hyperparameters; and the sliced variant is very fast to compute. However, we note that slicing has also been done for other distance measures, such as MMD with a specific choice of kernel (Hertrich et al., 2024) and mutual information (Goldfeld & Greenewald, 2021). We provide a formal definition of the Wasserstein distance below, and of the Sliced-Wasserstein distance in Appendix A.3.

**Definition of Wasserstein Distance**  We here provide the definition in the common case where probability density functions $p_1$ and $p_2$ are assumed to exist (formal definition using probability measures in Appendix A.3). Let $M \subseteq \mathbb{R}^d$, and $|| \cdot ||_q$ be the $q$-norm in $\mathbb{R}^d$. Then the Wasserstein-$q$ norm can be written as

$$W_q(p_1, p_2) = \inf_{\gamma \sim \Gamma(p_1, p_2)} \left( \mathbb{E}_{x_1, x_2 \sim \gamma} ||x_1 - x_2||_q^q \right)^{\frac{1}{q}}, \tag{1}$$

where $\Gamma(p_1, p_2)$ is the set of all couplings, that is all possible "transportation plans", between $p_1$ and $p_2$. $\gamma \in \Gamma(p_1, p_2)$ is a joint distribution over $(x_1, x_2)$ with respective marginals $p_1$ and $p_2$ over $x_1$ and $x_2$.

**Sample-Based Wasserstein Distance**  In practice, Eq. (1) is analytically solvable for only a few distributions. Therefore, Wasserstein distance is typically estimated from finite samples from $p_1$ and $p_2$. However, sample-based estimates of the Wasserstein distance are biased, and the convergence to the true Wasserstein distance is exponentially slower as the dimensionality of the distribution increases (Fournier & Guillin, 2015; Papp & Sherlock, 2022)

Intuitively, if two given probability distributions are thought of as two piles of dirt, the Wasserstein distance measures the (minimal) cost of "transporting" one pile of dirt to another (Panaretos & Zemel, 2019). This analogy led to the Wasserstein distance also being known as the Earth Mover's Distance. Suppose we have samples $\{x_1, \ldots, x_N\} \subset \mathbb{R}^d$ sampled from a distribution $p_1$ and $\{y_1, ..., y_N\} \subset \mathbb{R}^d$ sampled from another distribution $p_2$. Given any distance metric between two vectors in $\mathbb{R}^d$, $D(\cdot, \cdot)$, we can construct the *cost*

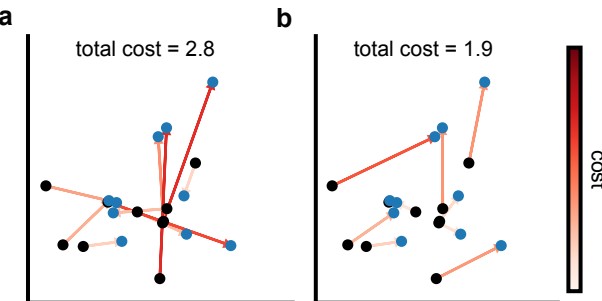

Figure 3: **Computing Wasserstein distance.** Two transport maps mapping the samples from a two-dimensional distribution $p_1$ (black) to samples from another distribution $p_2$ (blue), shown by arrows. The color of the arrow corresponds to the cost (Euclidean distance) between $x_i$ and $y_i$. **(a)** Randomly chosen transport map. **(b)** The optimal transport map, giving the smallest total cost. The total cost for the optimal map in (b) is the Wasserstein distance between these two sets of samples. Note that this schematic demonstrates transport maps for the *non-sliced* Wasserstein distance in two dimensions.

*matrix $C$*, as the matrix of pairwise distances between the samples $x_i$ and $y_j$:

$$C = \begin{bmatrix} D(x_1, y_1) & \dots & D(x_1, y_N) \\ \vdots & \ddots & \vdots \\ D(x_N, y_1) & \dots & D(x_N, y_N) \end{bmatrix} \tag{2}$$

Recalling the Earth Mover's Distance analogy, we want to map each $x_i$ to exactly one $y_j$, in such a way that the cost of doing so is minimized. The minimum transport map then defines the Wasserstein distance (for the metric $D$) between the two empirical distributions. Throughout this work, we employ the commonly used Euclidean metric, $L^2$, leading to the Wasserstein-2 and Sliced Wasserstein-2 distances. More precisely, we define a "transport map" to be a permutation matrix, $\pi \in \{0, 1\}^{N \times N}$, which is a matrix with exactly one nonzero entry in each row. The entry $\pi_{ij} = 1$ means that we transport the point $x_i$ to the point $y_j$. Then, finding the transport map that minimizes the overall cost can be stated as

$$\pi^* = \min_\pi \sum_{ij} \pi_{ij} C_{ij}. \tag{3}$$

A randomly chosen transport map for small datasets in $\mathbb{R}^2$ is shown in Fig. 3a. Fortunately, the optimal solution to Eq. (3) can be solved exactly using the *Hungarian method* (Kuhn, 1955), leading to the assignment shown in Fig. 3b.

**Slicing Wasserstein brings efficiency** Solving the optimal transport problem (Eq. (3)) with the Hungarian method has a time complexity of $O(N^3)$ in the number of samples $N$ (although faster $O(N^2 \log N)$ approximations exist, see Peyré et al. 2017). However, in the special case where the data is one-dimensional, the Wasserstein distance can be calculated by sorting the two datasets, obtaining the *order statistics* $\{x_{(1)}, .., x_{(N)}\}$ and $\{y_{(1)}, ..., y_{(N)}\}$ and computing the sum of the distances $\sum_i D(x_{(i)}, y_{(i)})$. This has a time complexity of $O(N \log N)$. Thus, slicing the Wasserstein distance with one-dimensional projections becomes very efficient. While the value of the SW distance does not converge to the Wasserstein distance, even in the case of an infinite number of data samples and slices, the SW distance is a metric (in the mathematical sense) as long as $D$ is a metric on $\mathbb{R}^d$, and it acts as a lower bound to the Wasserstein distance (Nadjahi, 2021).

**Limitations** When two high-dimensional distributions differ only along a small subset of directions, relying on one dimensional projections can become a significant limitation. This is because in high-dimensional spaces, most slices will be (almost) orthogonal to the few directions which distinguish the two distributions. In such a situation, the gain in efficiency from slicing can be impeded by the large number of slices we would need to reliably tell the two distributions apart. This limitation can be addressed by considering

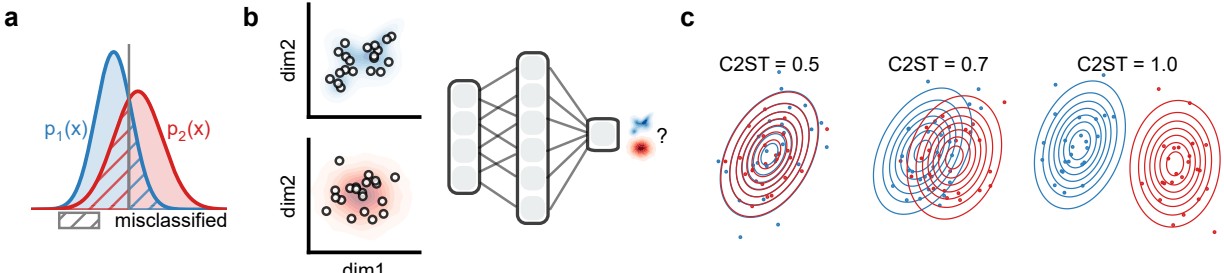

Figure 4: **Classifier Two-Sample Test (C2ST). (a)** The C2ST classifier problem: identifying the source distribution of a given sample. The optimal classifier predicts the higher-density distribution at every observed sample value, resulting in a majority of samples being correctly classified. **(b)** When probability densities of the distributions are not known, the optimal classifier is approximated by training a classifier, e.g., a neural network, to discriminate samples from the two distributions. **(c)** C2ST values vary from 0.5 when distributions exactly overlap (left) to 1.0 when distributions are completely separable (right).

nonlinear (Kolouri et al., 2019) or other specific (Deshpande et al., 2019; 2018) slices. Furthermore, slicing may also be relaxed to other kinds of data-specific projections, such as Fourier features for stationary time series or locally-connected projections for images (Du et al., 2023; Cazelles et al., 2021). Finally, while sliced distances can overcome computational challenges of computing distances in higher dimensions, they typically inherit the limitations of the distance being sliced.

## 2.2   Classifier-based: Classifier Two-Sample Test (C2ST)

The Classifier Two-Sample Test (C2ST) uses a classifier that discriminates between samples from two distributions (Fig. 4a; Lopez-Paz & Oquab 2017; Friedman 2003). The distance between the distributions can then be quantified by various measures of classifier performance. For example, one would train a classifier $c(x)$ to distinguish samples from the generative model and the data, and then evaluate the C2ST as $\frac{1}{2}[\mathbb{E}_{p(x)}[\mathbb{1}(c(x) = 0)] + \mathbb{E}_{q(x)}[\mathbb{1}(c(x) = 1)]]$. The classification accuracy provides a particularly intuitive and interpretable measure of the similarity of the distributions. If the classification accuracy is 0.5, i.e., the classifier is at chance level, the distributions are indistinguishable to the classifier (Fig. 4c, left), while higher accuracy indicate differences in the distributions (Fig. 4c, middle). If the C2ST is 1.0, the two distributions have no (or very little) overlap in their supports (Fig. 4c, right). Given two distributions, the C2ST has a 'true' (optimal) value, which is the maximum classification accuracy attainable by any classifier (Fig. 4a). This optimal value can be computed if both distributions allow evaluating their densities, but this is not usually possible if only data samples are available. In that case, one aims to train a classifier, such as a neural network (Fig. 4b), that is as close to the optimal classifier as possible.

One of the main benefits of the C2ST is that its value is highly interpretable (the accuracy of the classifier). C2ST can also be used to test the statistical significance of the difference between two sets of samples. However, calculating C2ST can be resource-intensive because it involves training a classifier. Additionally, using classifiers in a differentiable training objective can be challenging (but is sometimes possible, as in, e.g., Generative Adversarial Networks; Goodfellow et al. 2014). Furthermore, the value is dependent on the capacity of the classifier, and hence on many hyperparameters such as classifier architecture or training procedure. This dependence on a trained classifier can result in C2ST estimates that are biased, and the variety of possible classifier architectures means theoretical guarantees such as sample complexity are difficult to determine. In our experiments, we used a scikit-learn Multi-Layer Perceptron classifier, combined with a five-fold cross-validation routine to estimate the accuracy returned (Pedregosa et al., 2011).

**Common failure modes**   As mentioned above, for any realistic scenario the C2ST is computed by training a classifier. The resulting C2ST will only be a good measure of distance between real and generated data

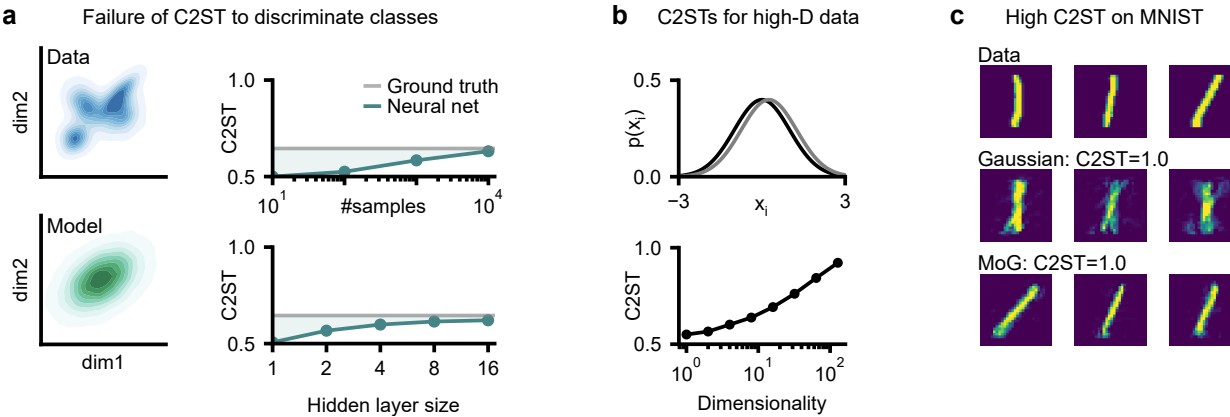

Figure 5: **Failure modes and behavior of C2ST. (a)** Data (top left) and Gaussian maximum-likelihood estimate (bottom left). C2ST wrongly returns 0.5 (no difference between the densities) if too few samples are used (top right) or the neural network is poorly chosen (bottom right). **(b)** For high-dimensional densities, despite the marginals between data (black) and model (gray) seeming well-aligned, small differences (here a mean shift of 0.25 std. in every dimension) allow the classifier to more easily distinguish the distributions as dimensionality increases, yielding correct but surprisingly high C2ST. **(c)** On MNIST, the C2ST between data (top) and a Gaussian generative model (middle) as well as of a Mixture of Gaussians (MoG, bottom) is 1.0, although the MoG is perceptually more aligned with the data.

if the classifier is close or equal to the optimal classifier. To demonstrate the behavior of the C2ST if this is not the case, we fitted a Gaussian distribution to data that was sampled from a Mixture of Gaussians (Fig. 5a, left). The optimal C2ST between these two distributions is 0.65 (which can be computed because Gaussians and Mixtures of Gaussians allow evaluating densities). If the C2ST is estimated with a neural network, however, we observe that this C2ST can be systematically underestimated: for example, when only few samples from the data and generative model are available, the neural network predicts a C2ST of 0.5 (Fig. 5a, top right)—in other words, it predicts that the generative model and the data follow the same distribution. Similarly, if the neural network is not expressive enough, e.g., with too few hidden units, the classifier will also return a low C2ST, around 0.5 (Fig. 5a, bottom right). These issues can make the C2ST easy to misuse. In many cases, reporting a low C2ST is desirable for generative models since it indicates that the model perfectly matches data, but one can achieve a low C2ST simply by not investing sufficient time into obtaining a strong classifier.

**C2ST can remain very high even for seemingly good generative models**   We previously argued that the C2ST is an interpretable measure—while this is generally true, the C2ST can sometimes be surprisingly high even if the generative model seems well aligned with the data. For example, when the generative model aligns very well with the data for every marginal, the C2ST can still be high if the data is high-dimensional (Fig. 5b). Because of this, it can be difficult to achieve low C2ST values on high-dimensional data. To further demonstrate this, we fitted a Gaussian distribution and a mixture of 20 Gaussian distributions to the 'ones' of the MNIST dataset. Although the Mixture of Gaussians (Fig. 5c, bottom row) looks better than a single Gaussian (Fig. 5c, middle row), both densities have a C2ST of 1.0 to the data (obtained with a ResNet on ≈4k held-out test datapoints).

**Other C2ST variants**   While we focus on a standard C2ST definition by using classification accuracy as the C2ST distance (Lopez-Paz & Oquab, 2017), any other performance measure for binary classification could be used (Raschka, 2014). Kim et al. (2019) argues that classic accuracy is sub-optimal due to the "binarization" of the class probabilities and proceeds to instead use the mean squared error between the predicted and "target" value of 0.5. Other approaches instead construct a likelihood ratio statistic (Pandeva et al., 2024).

Additionally, instead of using the estimated class probabilities, Cheng & Cloninger (2022) consider using the average difference in logits (i.e., activations in the last hidden layer).

We note that the learned classifier in C2ST can be applied to estimations of a density ratio $\frac{p(x)}{q(x)}$, also known as the likelihood ratio trick (Hastie et al., 2001; Sugiyama et al., 2012). Density ratio estimation has attracted a great deal of attention in the statistics and machine learning communities since it can be employed for estimating divergences between two distributions, such as the Kullback–Leibler divergence (Titsias & Ruiz, 2019; Huszár, 2017) and Pearson divergence (Srivastava et al., 2020).

### 2.3 Kernel-based: maximum mean discrepancy (MMD)

MMD is a popular distance metric that is applicable to a variety of data domains, including high-dimensional continuous data spaces, strings of text, and graphs (Borgwardt et al., 2006; Gretton et al., 2012a; Muandet et al., 2017). It has been used to evaluate generative models (Sutherland et al., 2021; Borji, 2019; Lueckmann et al., 2021) and also has the ability to indicate *where* the model and the true distribution differ (Lloyd & Ghahramani, 2015). The distance provided by MMD can straightforwardly be used to test whether the difference between two sets of high-dimensional samples is statistically significant (Gretton et al., 2012a).

To assess whether two sets of samples are drawn from the same distribution, MMD makes use of a kernel function to (implicitly) embed the samples via an embedding function $\phi$, also called a feature map. If we choose the right kernel, we can end up embedding our samples in a space where the properties of the underlying distributions are easily compared. We will motivate the use of the kernel in MMD by illustrating different explicit embeddings before introducing the implicit embedding via a kernel $k$. Note that this explanation is inspired by Sutherland (2019).

In a first step, we can define MMD as the difference between the means of the embedding of two distributions $p_1$ and $p_2$:

$$\mathsf{MMD}^2[\phi, p_1, p_2] = \|\mathbb{E}_{p_1(x)}[\phi(x)] - \mathbb{E}_{p_2(y)}[\phi(y)]\|^2,$$

for any embedding function $\phi$.

If we have samples of real numbers from two distributions $p_1$ and $p_2$ (Fig. 6a), there are various embedding functions $\phi$ we could use to compare these. The simplest possible function $\phi^{(1)} : \mathbb{R} \to \mathbb{R}$ is the identity mapping $\phi^{(1)}(x) = x$ (Fig 6b, left). However, in this case, the MMD will simply be the absolute difference between the means (first moments) of the distributions (for details, see Section A.2):

$$\mathsf{MMD}[\phi^{(1)}, p_1, p_2] = |\mu_{p_1} - \mu_{p_2}|.$$

This does not yet allow us to discriminate different distributions with equal means (all distributions in Fig. 6). If we now expand our embedding with a quadratic term, $\phi^{(2)} : \mathbb{R} \to \mathbb{R}^2$ as $\phi^{(2)}(x) = \begin{bmatrix} x \\ x^2 \end{bmatrix}$ (Fig 6b, right), the MMD yields (for details, see Section A.2)

$$\mathsf{MMD}^2[\phi^{(2)}, p_1, p_2] = (\mu_{p_1} - \mu_{p_2})^2 + (\mu_{p_1}^2 + \sigma_{p_1}^2 - \mu_{p_2}^2 - \sigma_{p_2}^2)^2.$$

In this case, we can also distinguish distributions with different variances (or second moments; allowing us to differentiate between two out of three distributions in Fig. 6). If we want to distinguish between all distinct distributions, we could keep adding additional features to $\phi$ to capture higher moments. However, this could become infeasible—if we want to make sure two probability distributions are exactly equal, i.e., have exactly the same moments, we would need to add infinitely many moments. Luckily, there is a trick we can exploit. First, we can rewrite MMD in terms of inner products of features (denoted with $\langle \cdot, \cdot \rangle$; for details, see Section A.2) as

$$\mathsf{MMD}^2[\phi, p_1, p_2] = \mathbb{E}_{p_1(x), p_1'(x')}[\langle \phi(x), \phi(x') \rangle] + \mathbb{E}_{p_2(y), p_2'(y')}[\langle \phi(y), \phi(y') \rangle] - 2\mathbb{E}_{p_1(x), p_2(y)}[\langle \phi(x), \phi(y) \rangle]$$

We can now rewrite the inner product $\langle \phi(x), \phi(x') \rangle$ in terms of a kernel function $k$: $\langle \phi(x), \phi(x') \rangle = k(x, x')$. Thus, if we can find a kernel for our feature map, we can avoid explicitly computing the features altogether and instead directly compute

$$\mathsf{MMD}^2[k, p_1, p_2] = \mathbb{E}_{p_1(x), p_1'(x')}[k(x, x')] + \mathbb{E}_{p_2(y), p_2'(y')}[k(y, y')] - 2\mathbb{E}_{p_1(x), p_2(y)}[k(x, y)].$$

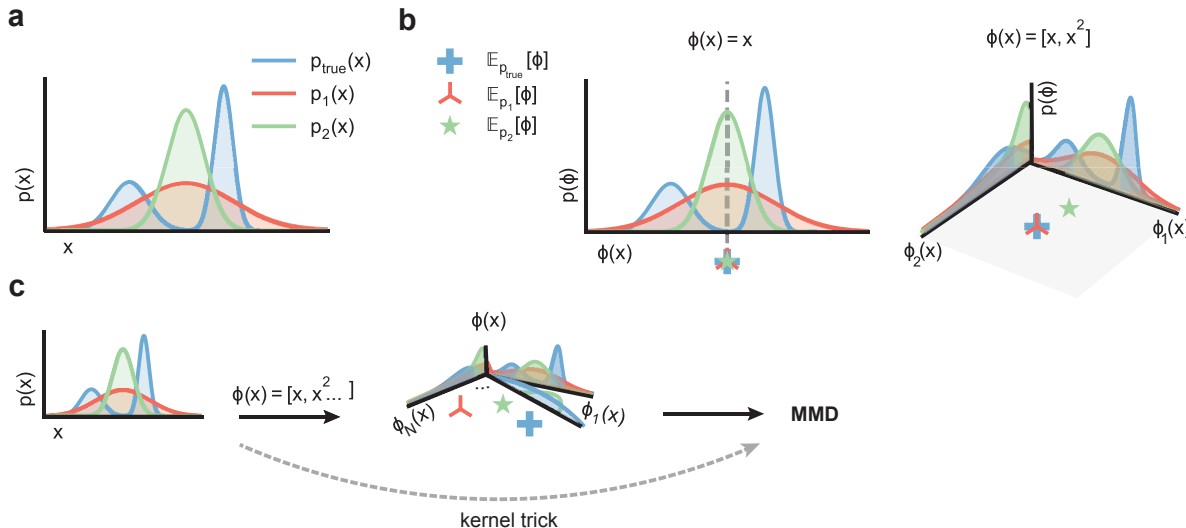

Figure 6: **Maximum mean discrepancy (MMD). (a)** Two example distributions $p_1(x)$, $p_2(x)$ and observed data $p_{true}(x)$ that we want to compare. **(b)** MMD can be defined as the difference between the expectations of some embedding function $\phi(x)$. If we take the identity as embedding ($\phi^{(1)}(x) = x$; left), we end up computing the differences between the means of the distributions, which are all equal for the three distributions. If we add a quadratic feature ($\phi^{(2)}(x) = [x, x^2]^\mathsf{T}$; right), we can distinguish distributions with different variances. Note that we still have $\mathsf{MMD}^2[\phi^{(2)}, p_2, p_{true}] = 0$, despite $p_2$ being different from $p_{true}$ **(c)** Using the *kernel trick* we can avoid computing the embeddings all together but use implicit embeddings that capture all relevant features of the distributions.

Evaluating the kernel function instead of explicitly calculating the features is often called the *kernel trick* (Fig. 6c). If we can define a kernel whose corresponding embedding captures all, potentially infinitely many moments, we would have an MMD that is zero only if two distributions are exactly equal and the MMD becomes a *metric*. These kernels are called *characteristic* (Section A.2, Gretton et al. 2012a), and include the commonly used Gaussian kernel: $k_G(x, x') = \exp(-\frac{\|x-x'\|^2}{2\sigma^2})$. A number of other kernels can also be used (see, e.g., Sriperumbudur et al. 2009). For instance, using a kernel induced by the Euclidean distance, MMD can be shown to be equivalent to the standard energy distance (Székely & Rizzo, 2013). In fact, a wider equivalence between MMD and the generalized energy distance has been established, using distance-induced kernels (Sejdinovic et al., 2013).

**MMD in practice**   Typically, the kernel version of MMD is used, which is straightforwardly estimated with its empirical, *unbiased*, estimate:

$$\mathsf{MMD}^2 = \frac{1}{m(m-1)} \sum_i^m \sum_{j \neq i} k(x_i, x_j) + \frac{1}{n(n-1)} \sum_i^n \sum_{j \neq i} k(y_i, y_j) - \frac{2}{mn} \sum_{i,j} k(x_i, y_j).$$

MMD in this form can be applied to many forms of data as long as we can define a kernel, which can include graphs (Vishwanathan et al., 2010; Gärtner, 2003) or strings of text (Lodhi et al., 2002), in addition to vectors and matrices.

When we estimate the MMD with a finite number of samples, the selection of the right kernel and its parameters becomes crucial. For example, when using a Gaussian kernel, one has to choose the bandwidth $\sigma$. The MMD approaches zero if we take $\sigma$ to be close to zero (then $k_G(x, x') = 1$ if $x = x'$ else $k_G(x, x') \to 0$) or if $\sigma$ is large (then $k_G(x, x') \to 1 \, \forall \, x, x'$; Gretton et al. 2012a). A common heuristic to remedy this parameter choice is picking the bandwidth based on the scale of the data. The *median heuristic* sets the bandwidth to the median distance between points in the aggregate sample (Gretton et al., 2012a). Another common approach is based on cross-validation, or data splitting (Gretton et al., 2012a;b; Jitkrittum et al., 2016;

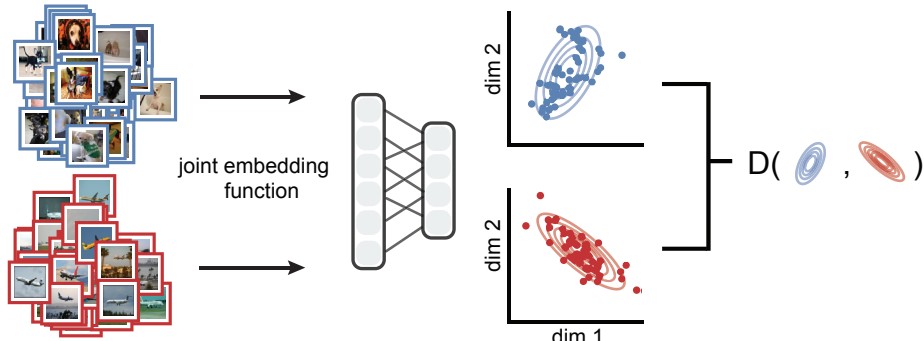

Figure 7: **Network-based metrics.** Instead of directly computing distances in data space, complex data, e.g., natural images of dogs sampled from $p_1(x)$ and aircraft sampled from $p_2(y)$, are jointly embedded into a vector space. The embedding function can, for example, be a deep neural network. The resulting distributions in feature space are then compared by a classical measure of choice $D$.

Sutherland et al., 2021): The dataset is divided, with a hold-out set used for kernel selection, and the other part used for evaluating MMD. While the data splitting method does not involve any heuristic, it can lead to errors in MMD since it reduces the number of data points available for estimating the MMD. Recent work attempts to choose hyperparameters without employing data splitting (Biggs et al., 2023; Schrab et al., 2023; Kübler et al., 2022b;a).

We often aim for a kernel that captures the (dis)similarity between the data points well, such a kernel can be domain-specific or specifically designed for downstream analysis tasks. The similarity between two strings (e.g., DNA sequences or text) can, for instance, be estimated by looking at the frequency of small subsequences (Leslie et al., 2001; Lodhi et al., 2002). Furthermore, it is possible to aggregate simpler kernels into a more expressive one (Gretton et al., 2012b), or to use a deep kernel (i.e., based on neural networks) that can exploit features of particular data modalities such as images (Liu et al., 2020; Gao et al., 2021).

## 2.4   Network-based: Embedding-space measures

Distribution comparisons on structured data spaces, such as a set of natural images, present unique challenges. Such data is usually high-dimensional (high-resolution images) and contains localized correlations. Furthermore, images of different object classes (such as airplanes and dogs) share low-level features in the form of edges and textural details but differ in semantic meaning. Similar challenges occur for time-series data, natural language text, and other complex data types (Smith & Smith, 2020; Jeha et al., 2021).

In this section, we rely on the example of natural images, but the presented framework generalizes to other data types. Naive distances would operate on a per-pixel basis, leading to scenarios where, for example, white dogs and black dogs are considered vastly different despite both being categorized as dogs. As we would like to have a distance measure that operates based on details relevant to the comparison, we can leverage neural networks trained on a large image dataset that captures features ranging from low-level to high-level semantic details: While earlier layers in a convolutional neural network focus on edge detection, color comparison, and texture detection, later layers learn to detect high-level features, such as a dog's nose or the wing of an airplane, which thought to be relevant for a meaningful comparison. Embedding-based distances use these activations of neural network layers as an embedding to compare the image distributions. The most popular distance in this class is the Fréchet Inception Distance (FID; Heusel et al. 2017), used to evaluate generative models for images. The FID uses a convolutional neural network's embeddings (specifically InceptionV3 (Szegedy et al., 2015)) to extract relevant features, applies a Gaussian approximation in the embedding space, and computes the Wasserstein distance on this approximation.

A FID-like measure, in essence, requires a suitable *embedding network* $f : \mathcal{X} \to \mathbb{R}^d$, where $f$ transforms the data from the original high-dimensional space $\mathcal{X}$ into a lower-dimensional, feature-rich representation in $\mathbb{R}^d$ (Fig. 7). Once the data samples are mapped into this reduced space through the embedding network, the two sets of embedded samples can be compared using the appropriate distance. When evaluating generative models for natural images, it is common to approximate the *embedded* distributions with Gaussian distributions by estimating their respective mean $\mu$ and covariances $\Sigma$. Under this Gaussian approximation, the squared Wasserstein distance (also known as the Fréchet distance) can be analytically computed as

$$W^2((\mu_1, \Sigma_1), (\mu_2, \Sigma_2)) = \|\mu_1 - \mu_2\|^2 + \text{Tr}\left(\Sigma_1 + \Sigma_2 - 2\left(\Sigma_1 \Sigma_2\right)^{\frac{1}{2}}\right). \tag{4}$$

In principle, any appropriate metric can be used in place of the Fréchet distance. For instance, Jayasumana et al. (2023); Bińkowski et al. (2018); Xu et al. (2018) use MMD as a metric in the embedding space, and the MMD-based Inception Distance is often referred to as Kernel Inception Distance (KID). KID is known to have some advantages over FID: unlike FID, KID has a simple, unbiased estimator with better sample efficiency (Jayasumana et al., 2023), and does not assume any parametric forms for the distributions. Since KID involves MMD, we must carefully select the proper kernel and its hyperparameter when applying it. A related and commonly used quality measure for images is the Inception Score (Salimans et al., 2016). In contrast to the FID, this measure uses the average InceptionV3 predicted class probabilities and compares them with the true marginal class distribution. Note that while both this score and the FID can agree with traditional distances (e.g., certain divergences), they might evaluate models differently (Betzalel et al., 2022); see Barratt & Sharma (2018) for further limitations of the Inception Score.

**Limitations**   One of the biggest limitations is the requirement of a suitable embedding network. Newer and more robust networks, such as the image network of the CLIP (Radford et al., 2021) vision-language model, provide better and more semantically consistent embeddings (Betzalel et al., 2022; Jayasumana et al., 2023) than the InceptionV3 network. However, as the embedding network is generally non-injective, identical distributions in the embedding space may not necessarily translate to identical distributions in the original space. Previous research has demonstrated the FID's sensitivity to preprocessing such as image resizing and compression (Parmar et al., 2022). Additionally, FID estimates are biased for finite sample sizes, making comparisons unreliable due to dependency on the generative model. However, methods to obtain a more unbiased estimate have been proposed ($\text{FID}_\infty$; Chong & Forsyth 2020; Betzalel et al. 2022).

## 3   Comparing distances: sample size, dimensionality, and more

In this section, we empirically compare these distances along several dimensions, with additional results presented in the Appendices. When evaluating (or training) generative models, it is important to understand that different statistical distances pay attention to different features of the generated samples. A key aspect to consider is that differences between distances can be especially pronounced in applications where we only have a limited amount of data points, e.g., identifying rare cell types (Marouf et al., 2020), or where we have very high-dimensional data, e.g., neural population recordings in neuroscience (Stringer et al., 2019). In such cases, one needs to ensure that the distance measures can reliably distinguish different distributions for the given sample set size while remaining computationally tractable. We therefore perform a number of empirical studies here: First, we investigate SW, C2ST and MMD when it comes to distinguishing data sets given varying numbers of samples and data with varying dimensionality (Section 3.1). Second, we investigate the scaling properties of FID on the ImageNet dataset (Section 3.2), as well as considering the other distances when computed on the InceptionV3 image embeddings. We also summarize the theoretical sample and computational complexities of the four discussed distances in Section A.6.

Furthermore, different distances might weigh differently the importance of sample diversity, or *mode coverage*, versus how well the modes of the true distribution are captured. If models only capture one mode, or a subset of the modes, i.e., exhibit *mode collapse*, they might have learned to generate realistic but unvaried samples. We illustrated the trade-off between mode collapse and mode coverage by optimizing mis-specified (Fig. S2) and well-specified (Fig. S3) models using the different distances (see also Section A.7 and Theis et al. 2016).

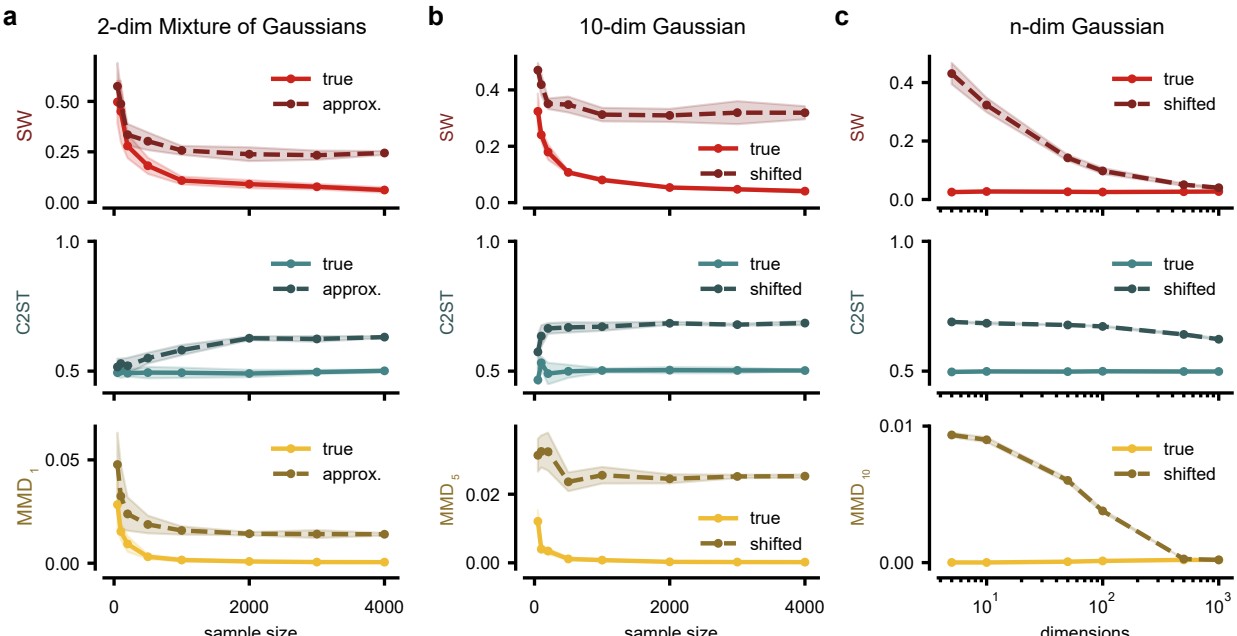

Figure 8: **Scalability of different statistical distances with sample size and dimensionality. (a,b)** Comparison of sample sets with varying sample size (between 50 and 4k samples per set) of a 'true' distribution, either with a second set of samples of the same distribution or with a sample set from a model consisting of an approximated/shifted distribution. We show the mean and standard deviation over five runs of randomly sampled data. Note that the subscript for MMD distances (bottom) denotes the bandwidth of the Gaussian kernel used for a given dataset and we report the squared distance for MMD. **(a)** Distances for the 2d-MoG example shown in Fig. 1 compared to samples from a uni-modal Gaussian approximation with the same mean and covariance. **(b)** Distances for a ten-dimensional standard normal distribution, with a model whose first dimension is shifted by one. **(c)** Distances based on 10k samples from a standard normal distribution with varying dimensions (between 5 and 1000). As in (b), *the first* dimension of the model is shifted by one. We show the mean and standard deviation over five runs of randomly sampled data. One MMD bandwidth was selected for all n-dimensional datasets.

Note that the absolute values of the distance measures can be hard to interpret and different measures lie on different scales. For this reason, we compare the distance between a set of sample from the model and a set of samples from the true distribution to the distance between two sets of samples from the true distribution.

### 3.1 Varying number of samples and dimensionality

Here, we assessed SW, C2ST and MMD using three datasets: First, we compared the two-dimensional Mixture of Gaussians ("2d-MoG") dataset introduced in Fig. 1 with samples from a model consisting of a unimodal Gaussian approximation with the same mean and covariance as the true data. Second, we compared samples from a *ten-dimensional* standard normal distribution with a model that is also a normal distribution that matches the true distribution in all but the first dimension—where the mean is shifted by one ("10-dim Gaussian"). Last, we repeated the second experiment, now taking a normal distribution with *varying dimensionality* as true dataset, and a model who again matches in all dimensions, except for the shifted first dimension ("n-dim Gaussian"). We used default parameters for the SW and C2ST measures while adjusting the bandwidth parameter for the MMD measure for each of the three comparisons.

**Sample size** We explored the robustness of the distances to low sample sizes on the 2d-MoG and the 10-dim Gaussian dataset. We found that for the 2d-MoG dataset, computed values quickly stabilized for

all measures by a sample size of 1000 and yielded the expected results of indicating low differences between two sets of samples from the true distribution (*true*), and a higher difference between samples from the true and model distributions (*approx.*). All measures failed to reflect the dissimilarity of the distributions at the lowest sample size of 50 samples (Fig. 8a). Here C2ST's behavior differs from MMD and SW, with C2ST indicating that the distributions are similar (C2ST $\approx 0.5$) while the other two distances indicate they are different (distance $\neq 0$).

For the 10-dim Gaussian dataset, we observed that all three distances can robustly identify samples from the same distribution as more similar than samples from different distributions with no measure being clearly superior to the others (Fig. 8b). For the previous 2d-MoG experiment, more samples were required to clearly detect the difference between the two distributions (Fig. 8a) as compared to the 10-dim Gaussian (Fig. 8b). Intuitively, the more pronounced the differences in the distributions we compare, the fewer samples we need to detect these differences (see additional experiments in Supp. Fig. S4).

**Dimensionality** We further tested how the distances scale with the data dimension using the n-dim Gaussian dataset. As the dimensionality increases, all distances indicate no difference between two sets of samples from the true distribution, except C2ST which is the only measure that consistently identifies samples from the true and model as distinct (Fig. 8c). When we changed the structure of the data distribution (e.g., by changing the mean of all dimensions or their variances, see Supp. Fig. S5), we observed a similar picture with some particularities: While the SW distance with a fixed number of slices has difficulties if the disparity between the distribution is only in one dimension, its performance drastically improves for differences in all dimensions, which is expected from the random projections SW is performing. C2ST seems to robustly detect differences even in high dimensions in these modified datasets, though previous experiments showed that this measure can be oversensitive to small changes in high dimensions (Fig. 5b,c). Lastly, while here MMD is not robust across different dimensions for a Gaussian kernel with *a fixed bandwidth*, one typically chooses the bandwidth to be on the order of the distance between datapoints (e.g., using the median heuristic; Section 2.3). Note that in general, MMD can be highly sensitive to kernel- and hyperparameter choice; an appropriate setting depends not only on the dimensionality of the data (Supp. Fig. S6, S7, and S8), but also on the structure of the distributions (Supp. Fig. S5). SW distance, on the other hand, appeared robust to number of random projections used (Supp. Fig. S1 and A.4).

### 3.2 FID-like distance comparison on ImageNet

To explore scaling properties of FID-like distances, we generated high-quality synthetic samples using a state-of-the-art diffusion model as described by Dockhorn et al. (2022), which we compared to images from the ImageNet dataset, which contains 1000 classes, 100 images per class. All images were embedded using the pre-trained InceptionV3 network (Deng et al., 2009; Szegedy et al., 2016), following the implementation of Heusel et al. (2017), transforming the raw images into a 2048-dimensional feature space.

**Sample size** We first produced varying number of samples with the base unconditional version of the diffusion model and compared those to samples from Imagenet (Fig. 9a). Calculating the FID involves computing the mean and covariance of the distributions in the embedding space and then calculating the squared Wasserstein distance analytically. However, we broadened our evaluation by applying additional distances to the distributions in the embedding space. While SW distance, C2ST, and FID effectively highlight the greater dissimilarity of synthetic samples to real images (i.e., the ImageNet test set) even for low sample sizes, the distinctiveness of the FID only becomes apparent when analyzing more than 2000 samples. This is in line with common implementations which generally recommend using at least 2048 samples (Heusel et al. 2018a;b). Another reason for the limited performance could be that the Gaussian assumption in the FID can be violated (Jayasumana et al., 2023; Betzalel et al., 2022). In contrast, the other distances reliably estimate a larger inter-dataset distance in regimes with few samples.

**Class discriminability** We aimed to determine the effectiveness of various distances in discriminating between images from different classes. To this end, we focused on comparing ImageNet images of dogs (D) with ImageNet images of other (non-dog) classes (~D; Huang et al. 2021). All investigated distances, except FID, were successful in identifying images from different classes as being more distinct than images from the

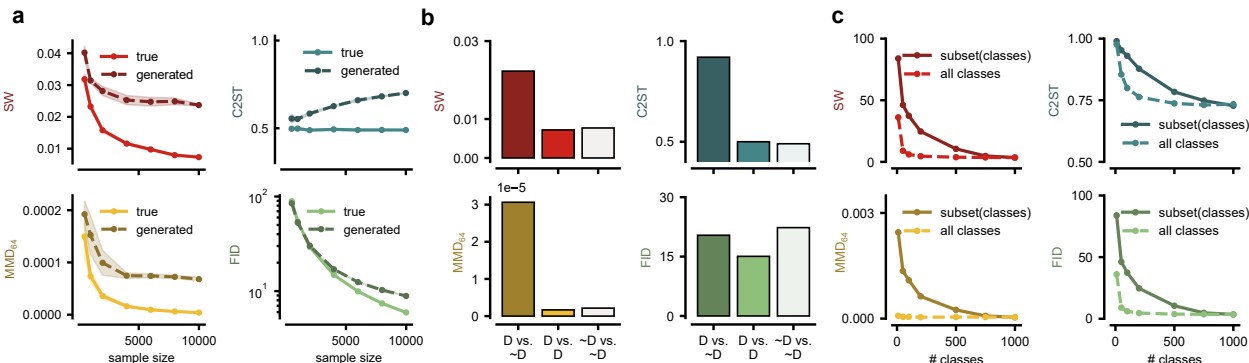

Figure 9: **Comparison of distances for ImageNet. (a)** A comparison between the ImageNet test set and samples generated by an unconditional diffusion model with varying sample sizes. **(b)** Distance evaluation on dog classes (D) versus non-dog classes (~D), highlighting differences in image representation between these two categories of real data. **(c)** Distances between sets of randomly selected images vs. varying number of included classes of images (from 10 to 1,000) from the test set, using synthetic samples created by a conditional diffusion model.

same class. The FID comparison between data with multiple classes (~D vs. ~D) is higher than across dog classes and other classes (D vs. ~D). This indicates that FID may be comparably insufficient in recognizing dissimilarities between distributions that have clear distinctions (whether across individual classes or in a one vs. all scenario).

**Mode coverage**  To examine the effects when only considering a subset of classes from the dataset (i.e., modes of the distribution), we used the diffusion model to generate class-conditional images matching the classses of the ImageNet test set (Fig. 9c). Our analysis involved comparing the complete test set against model-generated datasets that included only subsets of classes. For comparison, we created a control dataset by randomly sampling from all classes (dashed curve in Fig. 9c). This approach revealed that limiting the dataset to a small number of classes compromised the performance across all evaluated distances, in contrast to the outcomes observed when randomly excluding a subset of images—indicating that all distances successfully detected the lack of mode coverage. To achieve performance comparable to that observed with random removals, it was necessary to include at least 800 classes in the comparison. As the InceptionV3 network is, in essence, trained to classify ImageNet images (Szegedy et al. 2015; under certain regularization schemes), the extracted high-level features may also be very sensitive to class-dependent image features and not necessarily for general image quality. This behavior can be observed in Fig. 9c and was recently explored by Kynkäänniemi et al. (2023). By replacing InceptionV3 with other embedding networks (e.g., CLIP, which is trained to match images to corresponding text), this class sensitivity could be reduced (Kynkäänniemi et al., 2023).

**Additional generative models**  We additionally generated images using a consistency model (CM) for unconditional image generation (Song et al., 2023), to investigate how these metrics perform when evaluated on images created by different generative models. This model was trained on ImageNet 64x64, similar to the GENIE model (Dockhorn et al., 2022). Moreover, we included the following models: BigGAN (Brock et al., 2019), ablated diffusion model (ADM; Dhariwal & Nichol 2021), Glide (Nichol et al., 2022), Vector Quantized Diffusion Model (VQDM; Gu et al. 2022), Wukong (Wukong, 2022), Stable diffusion 1.5 (SD1.5; Rombach et al. 2022b) and Midjourney (Midjourney 2022; details in Appendix A.13). We evaluated the metrics (and multiple commonly used variants of KID and C2ST) for each model against the ImageNet test set (see Table S4).

While there is some agreement between the metrics regarding which model generates images closest to the ImageNet test set, there are also differences in the relative ordering across different metrics. As expected, the most recent unconditional models trained directly on ImagenNet 64x64 performed best in our

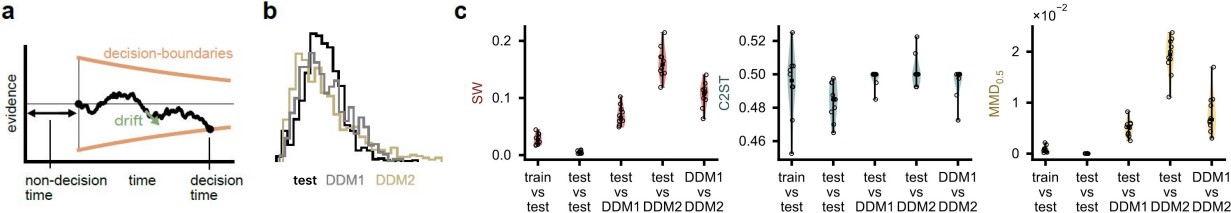

Figure 10: **Comparing models of primate decision making. (a)** Schematic of a Drift-Diffusion model (DDM), a classical neuroscientific model of decision making behavior. Overall, evidence drives the model toward one of two choices (drift), but sensory and environmental noise result in random fluctuations in evidence integration (diffusion). **(b)** Distributions of primate decision times from the test set (black), and two fitted models of varying complexity: DDM1 (gray) and DDM2 (gold). **(c)** SW distance, C2ST, MMD (bandwidth=0.5) between subsets of the generated and real data distributions. FID is not applicable in these comparisons, because the data are one-dimensional distributions. Scatter-points indicate comparisons between ten random subsets from each dataset. Thick horizontal bars indicate median values.

evaluation (GENIE, CS), better than the two other unconditional generative models (BigGAN, ADM). The other models are text-to-image and thus only prompted to generate images from specific ImageNet classes (GLIDE, VQDM, Wukong, SD1.5, Midjourney). Interestingly, the prompted models performed better than older unconditional models (BigGAN, ADM) most of the time. Recall that all we evaluate is the similarity to the ImageNet test set; prompted versions might produce images from the correct classes but might contain differences in style or appearance compared to actual images in ImageNet. Despite the demonstrated class-sensitivity (Fig. 9c), the InceptionV3 embeddings are thus also sensitive to different "styles" of natural images. Note that in this case, being closer to ImageNet does not necessarily mean generating better images (based on human perception), but rather creating images that are more ImageNet-like.

## 4 Scientific applications

To demonstrate how the presented distances apply to evaluating generative models of scientific applications, we focus here on two examples: decision modeling in cognitive neuroscience and medical imaging. For each application, we used two generative models or simulators to sample synthetic data. We then compared the synthetic samples to real data (hold-out test set) using the discussed distances. To obtain baseline values for each distance, we computed distances between subsets of real data. For SW distance, MMD, and FID we anticipated values proximal to zero, while for the C2ST, we expected a value around 0.5. These baseline assessments provide a lower threshold of model fidelity to which we compared the deviation of model-generated samples.

### 4.1 Models of primate decision making

We explored the fidelity of two generative models in replicating primate decision times during a motion-discrimination task (Roitman & Shadlen, 2002). We evaluated two versions of a Drift-Diffusion Model (DDM; Fig. 10a; Ratcliff 1978), a frequently used model in cognitive neuroscience. The two versions differ with respect to the drift rate, which is the speed and direction at which evidence accumulates towards a decision, and the decision boundaries, which determine how much evidence is needed to make a decision. Specifically, the first version (DDM1) uses a drift rate that varies linearly with position and time and decision boundaries that decay exponentially over time, whereas the second version (DDM2) uses a drift rate and decision boundaries that are constant over time (for details, see Section A.14). We fitted each model to empirical primate decision times with the use of the *pyDDM* toolbox (Shinn et al., 2020), generated one-dimensional synthetic datasets, and compared each dataset to the actual primate decision time distributions. While the resulting distributions of decision times are visually similar (Fig. 10b), the DDM-generated distributions DDM1 and DDM2 are noticeably broader compared to the more tightly clustered real decision times. Moreover, the

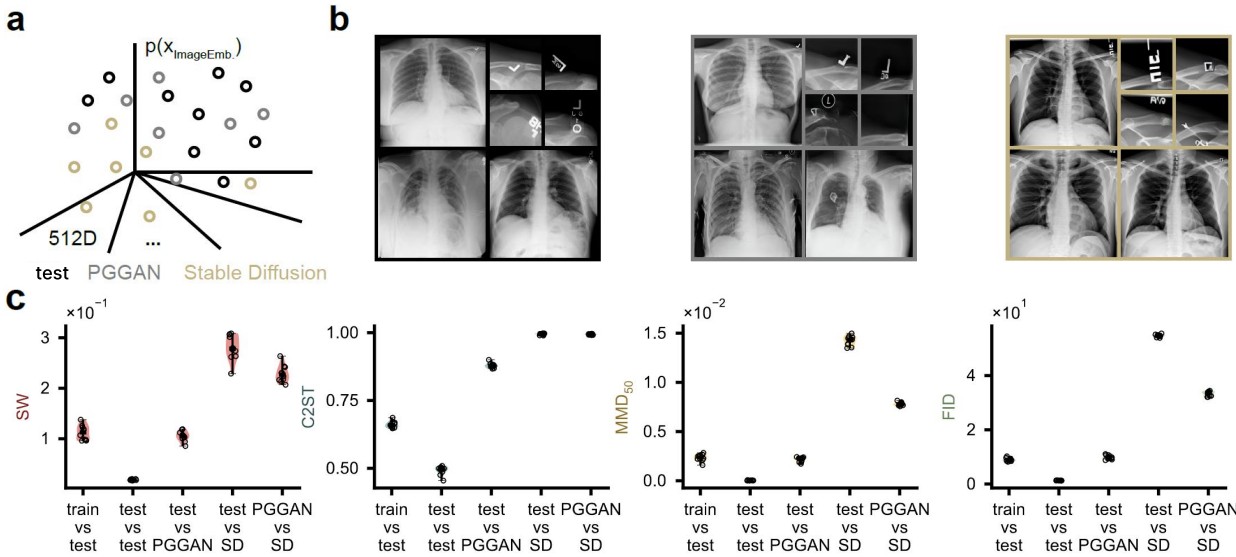

Figure 11: **Comparing generated and real X-ray images. (a)** Sketch of the embedded distributions of X-ray images from the three different datasets: test set of real dataset (black), Progressive Growing Generative Adversarial Network (PGGAN) (grey), and Stable Diffusion (SD) model (gold). **(b)** Examples from real and generated X-ray images. Three full-view examples from each distribution and four examples magnifying the top right corner. **(c)** SW distance, C2ST, MMD (bandwidth=50), and FID between samples of the real and generated distributions of embedded X-ray images. Scatter-points indicate comparisons between ten random subsets from each dataset. Thick horizontal bars indicate median values.

DDM1 distribution appears more similar to the real distribution than that of DDM2, which is shifted towards the left. As expected, the DDM1 model more precisely mimics the real data distribution, as compared to the DDM2, across the median values of the SW distance, MMD, and C2ST distances (Fig. 10c). For C2ST, DDM1 and the real data distribution are even indistinguishable, with median C2ST values around 0.5. This suggests that SW and MMD provide a more nuanced differentiation between the models.

## 4.2 Chest X-ray image generation

In the second application we turned to a high-dimensional example, in which we compared synthetic X-ray images generated by a Progressive Growing Generative Adversarial Network (PGGAN) model (Segal et al., 2021) and by a StableDiffusion (SD) (Malik & Humair, 2023) to real chest X-ray images from the ChestX-ray14 dataset (Wang et al., 2017). Each image has a total dimension of $1024 \times 1024$ pixels.

From visual inspection, we note two observations: First, the images produced by the SD model are clearer and sharper than either the real images or those generated by the PGGAN. Second, generated images contain unrealistic artifacts that distinguish them from real X-ray images (Fig. 11b). In real images, the top often contains annotations including, e.g., patient id, side of the body, or the date the X-ray was taken. These textual elements often contain artifacts or, in case of SD, are completely unrealistic. To compare these high-dimensional images, we embedded them in a 512-dimensional embedding space using the CheXzero network (Tiu et al., 2022), a CLIP (Radford et al., 2021) network fine-tuned for chest X-ray images. We opted for using this specialized network instead of the standard InceptionV3 network as it might overcome biases introduced by classification task training (Kynkäänniemi et al., 2023).

As expected, samples generated by PGGAN are closer to the real data across all distances compared to SD-generated data (Fig. 11c), likely due to unrealistic sharpness and more obvious textual artifacts of the SD-generated images.However, C2ST is even high between PGGAN outputs and the real data, suggesting

that the high-dimensionality of the data increases the sensitivity of this measure. Taken together, our results suggest that PGGAN is more accurate in generating realistic X-ray images compared to SD.

Our findings show that using different metrics can support different conclusions. For instance, C2ST suggests equality between DDM1 and real decision time data, whereas SW distance and MMD metrics indicate a larger difference between DDM1 and the real data. Similarly, in analyzing X-ray image generation, SW distance, MMD, and FID metrics suggest a high similarity between PGGAN-generated and real images, whereas C2ST indicates a strong difference. Thus, we want to highlight the importance of using multiple complementary distances for best results and understanding of model limitations.

## 5 Discussion

This work describes and characterizes four commonly applied sample-based distances representing different methodologies for defining statistical distance: Using low-dimensional projections (SW), obtaining a distance using classifiers (C2ST), using embeddings through kernels (MMD) or neural networks (FID). Despite their operational differences, they are all based on a fundamental concept: *simplifying complex distributions into more manageable feature representations to facilitate comparison.* Sliced distances effectively reduce multidimensional distributions to a set of one-dimensional distributions, where classical metrics are more easily applied or calculated. MMD uses kernels to (implicitly) project samples into a higher dimensional feature space, in which comparing mean values becomes more expressive. Classifier-based methods (C2ST) transform the task of distribution comparison into a classification problem; comparison is made by investigating how well a classifier can distinguish the distributions. Lastly, network-based distances, such as FID, explicitly map samples into a representative feature space and compare distributions directly within this space.

In the paragraphs below, we highlight the features and limitations of these investigated distances. Additionally, we discuss the relationships between these metrics and connect them to current related work.

**Sliced Distances** Sliced distances stand out for their computational efficiency in evaluating distributional discrepancies. However, when distributions differ primarily in lower-dimensional subspaces, sliced distances might not detect these subtle differences without a large number of slices (see Fig. 8c). There are approaches to reduce this effect by considering other projections than simple linear slices, as described in Section 2.1. In our experiments, the metric did show convincing results and in contrast to the MMD, C2ST, and FID, SW distance does not require one to choose specific hyperparameters (for which results can differ drastically). Although currently not extensively used in literature for evaluation, this makes the SW distance efficient, scalable, and a consistent baseline for general distribution comparisons.Yet, this also makes it less flexible to adapt to specific features of interest. The Wasserstein and SW distances are not interpretable, and admit only biased sample-based estimates. This can be a limitation for some tasks. However, due to its computational efficiency and differentiability, the SW distance is commonly used as a loss function to train generative models, such as GANs (Deshpande et al., 2018; Wu et al., 2019), Autoencoders (Wu et al., 2019), nonparametric flows (Liutkus et al., 2019), normalizing flows (Dai & Seljak, 2021), and multi-layer perceptrons (Vetter et al., 2024). Although the majority of research on sliced distances focuses on sliced *Wasserstein* metrics, slicing other metrics is also possible. For a certain subset of choices, equivalence to MMDs can be established (Feydy et al., 2019; Kolouri et al., 2019; Hertrich et al., 2024).

**Classifier Two-Sample Test (C2ST)** C2ST distinguishes itself by producing an interpretable value: classification accuracy. This characteristic makes C2ST particularly appealing for practical applications, as it is easy to explain and interpret. A notable drawback is the computational demand associated with training a classifier, which can be substantial. Moreover, C2ST's effectiveness is critically dependent on the selection and training of a suitable classifier. Interpreting results reported for C2ST requires knowledge of the classifier used and its appropriateness for the data at hand. Furthermore, automated training pipelines may encounter failures, such as when the trained classifier performs worse than chance, often due to overfitting to cross-validation folds (see also Section A.5). On the other hand, it is able to even detect subtle differences within two distributions in high dimensions. Even if there is a difference in only a single out of a thousand dimensions (for which SW distance and MMD might struggle), C2ST is able detect it (see Fig. 8c). This might be desirable, but can also be problematic. When comparing images, slight variations in a few pixels

may not be visually noticeable, potentially making them unimportant to the researcher. In high-dimensional complex data, such slight variations are quite likely. Thus C2ST can be close to 1.0 in the high-dimensional setting, making it practically useless for evaluation (see Fig. 5c, 11c). The C2ST can be shown to be a MMD with a specific kernel function parameterized by the classifier (Liu et al., 2020).

**MMD** The Maximum Mean Discrepancy is a strong tool for comparing two groups of data by looking at their average values in a special feature space. The effectiveness of MMD largely depends on the kernel function chosen (implicitly representing the feature space), which affects how well it can spot differences between various types of data. Inappropriate kernel choice can leave the metric insensitive to subtle differences in the distribution (Gretton et al. 2012b; Sriperumbudur et al. 2009; see Fig. 8). The MMD can be estimated efficiently and is differentiable, and thus often used as a loss function for training generative models (Dziugaite et al., 2015; Arbel et al., 2019; Li et al., 2017; Bińkowski et al., 2018; Briol et al., 2019). Yet, a kernel must satisfy certain criteria, e.g., positive definiteness, making the design of new kernel functions challenging. Such constraints are relaxed for FID-like metrics, which focus on *explicit* representations of the embedding, whereas (kernel) MMD instead focuses on *implicit* representations. One advantage, however, is that the implicit embedding allows for infinite dimensional feature spaces (through characteristic kernel functions). These can be proven to be able to discriminate *any* two distinct distributions, something that is impossible through explicit representations used by the FID. Recently, Kübler et al. (2022a) proposed a method to estimate MMD via a witness function that determines MMD (Appendix A.2). This method is closely related to C2ST in that both estimate a discrepancy among distributions via a classifier (Kübler et al., 2022a, Section 5).

**Network-based** Network-based approaches for evaluating distributions focus on the analysis of complex data, emphasizing the importance of capturing high-level, semantically meaningful features. These methods leverage neural networks to project data into a lower-dimensional, feature-rich space where traditional statistical distances can be applied more effectively. This is particularly important for tasks where the visual or semantic quality of the data is important, making them a popular choice for assessing generative models in domains such as image and text generation. The primary challenge lies in the design of suitable network architectures that can extract relevant features for accurate distribution comparison. Even more important than for the C2ST, this network must be well-established and shared which is a necessary but not sufficient criterion (Chong & Forsyth, 2020) to compare different results. While such well-established defaults exist for images (Szegedy et al., 2015; Radford et al., 2021), this is not the case for other domains. For example, the time series generation community did not yet establish a default, and embedding networks are either trained or chosen by the authors (Smith & Smith, 2020; Jeha et al., 2021). We demonstrated that the class-sensitivity of FID (Section 3.2) often leads to model collapse, as seen in GANs. However, it may not accurately reflect the overall image quality. For example, Betzalel et al. (2022); Kynkäänniemi et al. (2023); Jayasumana et al. (2023) found that relevant features sometimes can disagree with human judgment and that CLIP embeddings align more closely to what humans perceive as favorable or unfavorable. Yet, FID features have been shown to align much better with human perception than traditional metrics (Zhang et al., 2018). Recent developments in network-based approaches include the use of Central Kernel Alignment (CKA; Cortes et al. 2012) to compute the distance between network-embedded samples. CKA scores show considerable stability when evaluated with different choices of network architectures and layers (Yang et al., 2023). Another newly introduced metric, Mauve (Pillutla et al., 2021; Liu et al., 2021; Pillutla et al., 2023), can, for instance, be used to measure how close machine-generated text is to human language using an external language model to embed the samples from each distribution. This metric uses divergence frontiers to take into account the trade-off between quality and diversity when evaluating generative models.

**Closing remarks** Ultimately, the choice of distance hinges on the nature of the data under consideration and the specific characteristics one aims to compare. Given a particular dataset and problem, it may be necessary to look beyond the distances discussed in this paper. For instance, in the realm of human-centric data such as images and audio, perceptual indistinguishability of distributions is crucial (Gerhard et al., 2013; Zhang et al., 2018). Time series data, with its inherent temporal structure, requires metrics that accommodate temporal shifts and variations without disproportionately penalizing minor discrepancies in timing, such as Dynamic Time Warping (Müller, 2007) or frequency-based methods (Hess et al., 2023). Additional recent works propose new distances based on fields such as topology (Barannikov et al. 2021).

In general, regardless of the specific use case, it is advisable to use multiple distance measures to obtain a comprehensive view, as relying on a single measure may lead to competing conclusions about the model under evaluation.

Throughout this paper, we have explained and analyzed four approaches to measuring statistical distance. While this represents only a small subset of all possible measures, we hope to have provided the foundational knowledge researchers need to find, understand, and interpret statistical distances relevant to their own scientific applications.

### Code availability

All code for replicating and running our analysis is available at: https://github.com/mackelab/labproject. Note that for implementations of the C2ST we largely made use of those from the simulation-based-inference package (Tejero-Cantero et al., 2020).

### Author Contributions

Conceptualization: AD, AS, CS, GM, JKL, LH, MD, MG, MP, SB, ZS
Software: AD, AES, AS, FG, FP, GM, JK, JKL, JV, LH, MD, MG, MP, RG, SB, ZS
Writing—Original Draft: AS, CS, FG, FP, GM, JK, JV, LH, MD, MG, MP, RG, RR, SB, ZS
Writing—Review & Editing: AD, AS, CS, FP, GM, JHM, JKL, JV, LH, LU, MG, MP, RG, SB, ST, ZS
Visualization: AD, AES, AS, FP, GM, JKL, JV, MD, MG, MP, RG, RR, ZS
Project administration: AS, MG, MP, SB
Funding acquisition: JHM

### Acknowledgments

We thank Eiki Shimizu and Nastya Krouglova for reviewing an early version of this paper. This project was inspired by a similarly structured project by the research group led by Karim Jerbi.

This work was supported by the German Research Foundation (DFG) through Germany's Excellence Strategy (EXC-Number 2064/1, PN 390727645) and SFB1233 (PN 276693517), SFB 1089 (PN 227953431), SPP2041 (PN 34721065), the German Federal Ministry of Education and Research (Tübingen AI Center, FKZ: 01IS18039; DeepHumanVision, FKZ: 031L0197B; Simalesam: FKZ 01|S21055A-B), the Carl Zeiss Foundation (Certification and Foundations of Safe Machine Learning Systems in Healthcare), the Else Kröner Fresenius Stiftung (Project ClinbrAIn), and the European Union (ERC, DeepCoMechTome, 101089288). ST was partially supported by Grant-in-Aid for Research Activity Start-up 23K19966, Grant-in-Aid for Early-Career Scientists 24K20750 and JST CREST JPMJCR2015. SB, MD, MG, JK, JL, GM, MP, RR, AS, ZS and JV are members of the International Max Planck Research School for Intelligent Systems (IMPRS-IS).

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

# A  Appendix

## A.1  Generative Models in Science

Table S1: **Example generative models in science.** This is not an exhaustive list regarding disciplines using generative models nor generative models used in the listed disciplines.

| Discipline | Model |
|---|---|
| Biology | |
| - Neuroscience | [1–13] |
| - single cell sequencing | [14–24] |
| - cellular biology [25] | [26–32] |
| Geoscience | |
| - ice flow modelling | [33–40] |
| - Numerical weather prediction | [41–46] |
| Chemistry | |
| - molecule generation [47] | [48–58] |
| Astronomy | |
| - astronomical images | [59–63] |

## A.2  Details about Maximum Mean Discrepancy

Here we provide different formulations and examples of MMD.

**Definition A.1 (Feature Map Definition of MMD)**

$$\text{MMD}^2[\phi, p_1, p_2] = \|\mathbb{E}_{p_1(x)}[\phi(x)] - \mathbb{E}_{p_2(y)}[\phi(y)]\|_{\mathcal{H}}^2, \tag{5}$$

*where $p_1(x)$ and $p_2(y)$ are the probability distributions of random variables $x, y \in \mathcal{X}$, and $\phi : \mathcal{X} \to \mathcal{H}$.*

Generally, $\mathcal{X}$ and $\mathcal{H}$ are defined as a topological space, and the reproducing kernel Hilbert space (RKHS), respectively, but readers can simply think of the Euclidean space $\mathbb{R}^N$ for the first examples in the main text.

For the *identity feature map* $\phi^{(1)} : \mathbb{R} \to \mathbb{R}, \phi^{(1)}(x) = x$, MMD can be computed as

$$\text{MMD}^2[\phi^{(1)}, p_1, p_2] = \|\mathbb{E}_{p_1(x)}[x] - \mathbb{E}_{p_2(y)}[y]\|_{\mathbb{R}}^2$$
$$= (\mathbb{E}_{p_1(x)}[x] - \mathbb{E}_{p_2(y)}[y])^2$$
$$\text{MMD}[\phi^{(1)}, p_1, p_2] = |\mu_{p_1} - \mu_{p_2}|.$$

And for the *quadratic polynomial feature map* $\phi^{(2)} : \mathbb{R} \to \mathbb{R}^2, \phi^{(2)}(x) = \begin{bmatrix} x \\ x^2 \end{bmatrix}$, MMD can be computed as

$$\text{MMD}^2[\phi^{(2)}, p_1, p_2] = \|\mathbb{E}_{p_1(x)}\left[\begin{bmatrix} x \\ x^2 \end{bmatrix}\right] - \mathbb{E}_{p_2(y)}\left[\begin{bmatrix} y \\ y^2 \end{bmatrix}\right]\|_{\mathbb{R}}^2$$

$$= \|\begin{bmatrix} \mu_{p_1} \\ \mu_{p_1}^2 + \sigma_{p_1}^2 \end{bmatrix} - \begin{bmatrix} \mu_{p_2} \\ \mu_{p_2}^2 + \sigma_{p_2}^2 \end{bmatrix}\|_{\mathbb{R}}^2$$

$$\text{MMD}^2[\phi^{(2)}, p_1, p_2] = (\mu_{p_1} - \mu_{p_2})^2 + (\mu_{p_1}^2 + \sigma_{p_1}^2 - \mu_{p_2}^2 - \sigma_{p_2}^2)^2.$$

**Definition A.2 (Kernel Definition of MMD)**

$$\text{MMD}^2[\phi, p_1, p_2] = \|\mathbb{E}_{p_1(x)}[\phi(x)] - \mathbb{E}_{p_2(y)}[\phi(y)]\|_{\mathcal{H}}^2$$
$$= \langle \mathbb{E}_{p_1(x)}[\phi(x)], \mathbb{E}_{p_1(x)}[\phi(x)]\rangle_{\mathcal{H}} + \langle \mathbb{E}_{p_2(y)}[\phi(y)], \mathbb{E}_{p_2(y)}[\phi(y)]\rangle_{\mathcal{H}} - 2\langle \mathbb{E}_{p_1(x)}[\phi(x)], \mathbb{E}_{p_2(y)}[\phi(y)]\rangle_{\mathcal{H}}$$
$$= \mathbb{E}_{p_1(x),p_1'(x')}[\langle \phi(x), \phi(x')\rangle_{\mathcal{H}}] + \mathbb{E}_{p_2(y),p_2'(y')}[\langle \phi(y), \phi(y')\rangle_{\mathcal{H}}] - 2\mathbb{E}_{p_1(x),p_2(y)}[\langle \phi(x), \phi(y)\rangle_{\mathcal{H}}]$$
$$\text{MMD}^2[k, p_1, p_2] = \mathbb{E}_{p_1(x),p_1'(x')}[k(x, x')] + \mathbb{E}_{p_2(y),p_2'(y')}[k(y, y')] - 2\mathbb{E}_{p_1(x),p_2(y)}[k(x, y)].$$

The definition of MMD can be rewritten through the notion *kernel mean embedding*. For given distribution $p(x)$, the kernel mean embedding $\mathbb{E}_{p(x)}[k(x, u)] \in \mathcal{H}$ is an element in RKHS that satisfies $\langle \mathbb{E}_{p(x)}[k(x, u)], f(u) \rangle_{\mathcal{H}} = \mathbb{E}_{p(x)}[f(x)]$ for any $f \in \mathcal{H}$ with argument $u \in \mathcal{X}$. The embedding $\mathbb{E}_{p(x)}[k(x, u)]$ is known to be determined uniquely if a corresponding kernel is bounded, *i.e.,* $\|k(x, x')\|_{\mathcal{H}} < \infty$ for any $x$. Then, as shown in Gretton et al. (2012a), $\mathsf{MMD}^2$ can be represented as

$$\mathsf{MMD}^2[k, p_1, p_2] = \|\mathbb{E}_{p_1(x)}[k(x, u)] - \mathbb{E}_{p_2(y)}[k(y, u)]\|_{\mathcal{H}}^2.$$

MMD can also be defined more generally as the integral probability metric.

**Definition A.3 (Supremum Definition of MMD)**

$$\mathsf{MMD}[\mathcal{F}, p_1, p_2] = \sup_{f \in \mathcal{F}} (\mathbb{E}_{p_1(x)}[f(x)] - \mathbb{E}_{p_2(y)}[f(y)]). \tag{6}$$

*Here, $\mathcal{F}$ is a class of functions $f : \mathcal{X} \to \mathbb{R}$. Where we take $\mathcal{F}$ as the unit ball in an RKHS $\mathcal{H}$ with associated kernel $k(x, x')$ (Gretton et al., 2012a), the function that attains supremum (the witness function) is*

$$f(u) = \frac{\mathbb{E}_{p_1(x)}[k(x, u)] - \mathbb{E}_{p_2(y)}[k(y, u)]}{\|\mathbb{E}_{p_1(x)}[k(x, u)] - \mathbb{E}_{p_2(y)}[k(y, u)]\|_{\mathcal{H}}}.$$

Assigning $f(u)$ into (Eq. (6)), we have

$$\begin{aligned}
\mathsf{MMD}^2[\mathcal{F}, p_1, p_2] &= \left( \sup_{f \in \mathcal{F}} (\mathbb{E}_{p_1(x')}[f(x')] - \mathbb{E}_{p_2(y')}[f(y')]) \right)^2 \\
&= \left( \frac{\mathbb{E}_{p_1(x), p_1'(x')}[k(x, x')] + \mathbb{E}_{p_2(y), p_2'(y')}[k(y, y')] - 2\mathbb{E}_{p_1(x), p_2(y)}[k(x, y)]}{\|\mathbb{E}_{p(x)}[k(x, u)] - \mathbb{E}_{p_2(y)}[k(y, u)]\|_{\mathcal{H}}} \right)^2 \\
&= \left( \frac{\|\mathbb{E}_{p_1(x)}[k(x, u)] - \mathbb{E}_{p_2(y)}[k(y, u)]\|_{\mathcal{H}}^2}{\|\mathbb{E}_{p_1(x)}[k(x, u)] - \mathbb{E}_{p_2(y)}[k(y, u)]\|_{\mathcal{H}}} \right)^2 \\
&= \|\mathbb{E}_{p_1(x)}[k(x, u)] - \mathbb{E}_{p_2(y)}[k(y, u)]\|_{\mathcal{H}}^2,
\end{aligned}$$

which is equal to the kernel definition of MMD.

**Definition A.4 (Characteristic Kernel)** *A kernel is called* characteristic *when the kernel mean embedding*

$$p(x) \mapsto f(u) = \mathbb{E}_{p(x)}[k(x, u)] \in \mathcal{H}$$

*is injective (Sriperumbudur et al., 2011; Fukumizu et al., 2008).*

This means that, if a characteristic kernel is used, the embedding into the RKHS can uniquely preserve all information about a distribution. In our evaluation, we utilize the Gaussian kernel, one of the well-known characteristic kernels. Another example of a characteristic kernel is the Laplacian kernel, which is defined by

$$k(x, x') := \exp\left( - \beta |x - x'| \right).$$

Note that linear and polynomial kernels are not characteristic, while they are quite popular in natural language processing.

### A.3 Formal Definition of Wasserstein and Sliced-Wasserstein Distance

Let $(M, \rho)$ be a Polish space, $\mu, \nu \in P(M)$ be two probability measures, and let $q \in [1, +\infty)$. Then the Wasserstein-$q$ distance between $\mu$ and $\nu$ is defined as

$$W_q(\mu, \nu) = \left( \inf_{\gamma \in \Gamma(\mu, \nu)} \mathbb{E}_{x_1, x_2 \sim \gamma} \rho(x, y)^q \right)^{1/q},$$

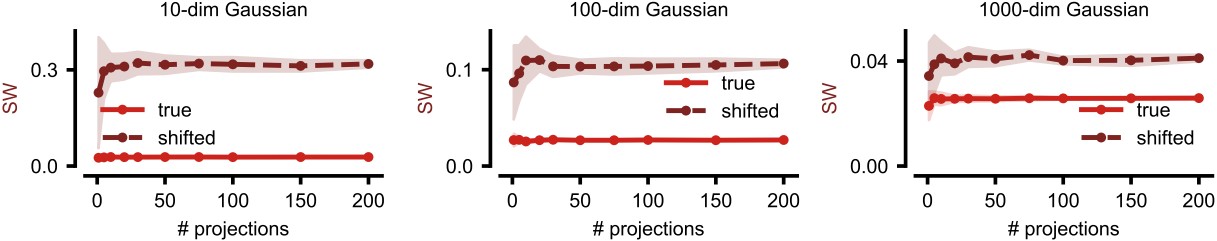

Figure S1: **The SW distance estimate is not strongly sensitive to the number of random projections.** We compare the SW distance estimate for the $\{10, 100, 1000\}$-dimensional Gaussian task with 1 shifted dimension (Sec. 3) as we increase the number of random projections used in the estimation. As the number of projections increases, the variance of the SW distance estimate decreases, but across all dimensionalities considered, the SW distance estimate has converged by 100 random projections.

where $\Gamma(\mu, \nu)$ is the set of all couplings between $\mu, \nu$.

The Sliced-Wasserstein distance is closely connected to the Radon transform (Helgason, 1980). We direct readers to Bonneel et al. (2015) for details. Here we provide a shorter definition following Definition 2.9 of Nadjahi (2021).

Suppose that $M \subset \mathbb{R}^d$, and denote by $\mathbb{S}^{d-1} = \{\theta \in \mathbb{R}^d : ||\theta||_2 = 1\}$ be the unit sphere with respect to the Euclidean norm. Let $u^* : X \to \mathbb{R}$ be a the linear form given by $u^*(x) = \langle u, x \rangle$, and $q \in [1, +\infty)$. Then the Sliced-Wasserstein distance (of order $q$) is defined for any measures $\mu, \nu \in P_q(X)$ as

$$SW_q(\mu, \nu) = \left( \int_{\mathbb{S}^{d-1}} W_q^q(u_\#^* \mu, u_\#^* \nu) d\mathcal{U}(u) \right)^{1/q},$$

Where $\mathcal{U}$ is the uniform distribution on $\mathbb{S}^{d-1}$, and for any $u \in \mathbb{S}^{d-1}$, $u_\#^*$ denotes the push-forward operator of $u^*$.

As for the Wasserstein distance, the definition becomes more intuitive in the case where $\mu$ and $\nu$ admit the probability density functions ($p_1$ and $p_2$ respectively). In particular, the random projection $u$ are uniformly random vectors in $\mathbb{S}^{d-1}$. Therefore, projecting the distributions $p_1$ and $p_2$ onto $u$ induces one-dimensional distributions $p_1^u$ and $p_2^u$ with samples $u^\mathsf{T} x_i$, where $x_1 \sim p_1$ and $x_2 \sim p_2$. The Sliced-Wasserstein-$q$ distance can then be written as

$$SW_q(p_1, p_2) = \left( \mathbb{E}_{u \sim \mathcal{U}(\mathbb{S}^{d-1})} [W_q^q(p_1^u, p_2^u)] \right)^{1/q}. \tag{7}$$

### A.4 Dependence of SW Distance on Number of Projections

All SW distance experiments in the main text were performed with the SW distance approximate with 100 random projections to approximate the expectation Eq. (7). Here, we additionally show the dependence of the SW distance approximation with finite projections on the $d$-dim Gaussian example (see Sec. $\pm$3), for $d \in \{10, 100, 1000\}$ (Fig S1). While the sample-based approximation to the analytic 1-dimensional Wasserstein distance is biased (Sec. 2.1), the Monte Carlo approximation to the expectation Eq. (7) is an unbiased estimate of the sample-based Wasserstein distance. We observe that for very high (1000)-dimensional distributions, the SW distance estimate converges quickly, and the choice of 100 random projections for the computation of the SW distance is appropriate.

### A.5 C2ST scores below 0.5

In practice the C2ST score can sometimes turn out to be below .5. That is, the trained classifier performs systematically worse than a random classifier. A potential reason for this effect is the existence of near duplicates or copies between the two different sets. Before training the classifier, these duplicates are assigned

Table S2: **Summary of computational complexity and theoretical properties of metrics in terms of number of samples $N$ and data dimensionality $D$.** Sample complexity here refers to the convergence rate of the sample-based estimate to the true value of the metric. *Bound based on Ghosal & Sen (2019); Nguyen & Ho (2024) for SWD and Gretton et al. (2012a) for MMD. †Best case scenario. In practice, the computational cost of training a neural network scales superlinearly with both sample size and data dimensionality. ‡ General case (see Sec. 2.3, A.6 for details).*Cost of calculating the square root of the covariance matrix in Eq. 4.

|  | SW | C2ST | MMD | FID |
|---|---|---|---|---|
| Sample Complexity ($N$) | $\mathcal{O}(N^{-1/2})$* | N/A | $\mathcal{O}(N^{-1/2})$* | N/A |
| Computational Complexity ($N$) | $\mathcal{O}(N \log N)$ | $\mathcal{O}(N)^{\dagger}$ | $O(N^2)^{\ddagger}$ | $\mathcal{O}(N^2)$ |
| Computational Complexity ($D$) | $\mathcal{O}(D)$ | $\mathcal{O}(D)^{\dagger}$ | $\mathcal{O}(D)$ | $\mathcal{O}(D^3)$* |
| Estimator unbiased? | no | no | yes | no |

opposite class labels. When a given pair of such duplicates is then split into one that belongs to the training set and one that belongs to the test set, the classifier is biased towards predicting the wrong class for the duplicate in the test set. This effect is particularly noticeable if the classifier was not carefully regularized during training, and thus memorized the class label of the duplicate in the training set.

## A.6 Sample and computational complexity

We here highlight that the different distances also differ in their sample and computational complexities (Table S2). With respect to the number of samples $N$, MMD and FID have a complexity of $O(N^2)$. Note that for MMD the computational cost can be reduced, e.g., by increasing variance or for a specific kernel choice (Gretton et al., 2012a; Zhao & Meng, 2015; Cheng & Xie, 2021; Bodenham & Kawahara, 2023; Bharti et al., 2023; Gretton et al., 2012b)). While SW distance scales with $O(N \log N)$ (Nadjahi, 2021), it is difficult to make principled assessments of the computational complexity for C2ST, as it is highly dependent on the chosen classifier. However, as more samples lead to larger training and test datasets, the sample size is likely to influence the compute time. Similarly, the computational complexity of computing these distances increases as the dimensionality of the data increases, with non-trivial scaling depending on the task and chosen hyperparameter. We also report the theoretical convergence of sample-based estimates for MMD, as well as SW, which is subject to active research. We report bounds from recent works (Ghosal & Sen, 2019; Gretton et al., 2012a). We do not report sample complexities for C2ST and FID, as these strongly depend on the choice of classifier and embedding network, respectively.

Note that despite their differences, all presented distances are reasonably tractable in the settings of our experiments, whereas the scaling experiments might be computationally unfeasible for other distances or datasets. Therefore, we strongly recommend carefully considering the complexity of the measure before conducting experiments on high-dimensional or very large datasets. We report the practical computation times for our experiments in Appendix A.11.

## A.7 Mode coverage properties

Mode coverage is the ability of a model to capture and generate diverse data, i.e., from multiple modes of the underlying distribution (see Fig. S2; S3). The community has focused on evaluation of mode coverage with different metrics driven by the development of GANs (Goodfellow et al., 2014; Gui et al., 2023; Saad et al., 2024). Empirically, in training generative models, mode coverage has been found to trade off with sample quality and speed, illustrating the generative learning trilemma (Xiao et al., 2022). All presented metrics can distinguish between a collapsed and a full distribution (Fig. 9), an empirical finding also reported in previous work (Che et al., 2017; Li et al., 2017; Deshpande et al., 2018; Borji, 2019). However, their sensitivity relies on different factors: SWD captures different modes when they are separated in one-dimensional projections, MMD depends on appropriate kernel choice, FID on expressive embeddings, and C2ST on well-trained classifiers. Other metrics used to quantify mode collapse include precision and recall (Kynkäänniemi et al., 2019).

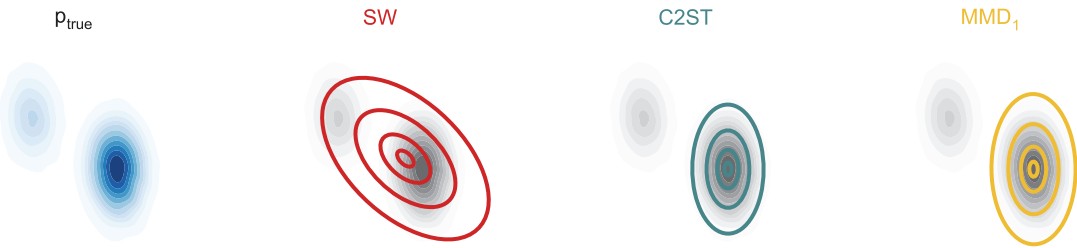

Figure S2: **Trade-offs illustrated through optimization of a miss-specified model.** We fitted a misspecified model (a two-dimensional Gaussian) by using different distances to a multi-modal distribution (similar to Theis et al. 2016). Note, that for a well-specified model each distance would give a perfect fit (Section A.8.2). In the optimization we minimized the SW distance, the C2ST classifiability, and the MMD with a Gaussian Kernel (details in Section A.8.1). Plotted are the contour lines of .25, .75, 1, and 2.5 standard deviations of the fitted Gaussians. The model optimised with SW is *mass-covering*: it covers both modes and therefore also assigns density to low-density regions of the true distribution, thus producing varied, but potentially unlikely samples. The models optimised with C2ST and MMD are *mode-seeking*: they have high densities only in the largest mode of the true distribution, and thus produces likely, but unvaried samples.

### A.8 Sliced-Wasserstein, MMD and C2ST as optimization target

#### A.8.1 Fitting a Gaussian with gradient descent

We provide an illustrative example of which distributions are obtained when using Wasserstein, MMD and C2ST distances as a goodness of fit criterion, for both a miss-specified example, in Fig. S2 and well-specified example Fig. S3. We can see that in the miss-specified example different distances make different trade-offs, for example whether they are mode-seeking, and produce likely but unvaried samples, or are mode-covering, where they produce varied, but also potentially unlikely samples.

For Wasserstein, optimisation has been studied more formally in previous work (Bernton et al., 2019; Yi & Liu, 2023). Wasserstein was used as training objective in Arjovsky et al. (2017) and MMD was used as training objective in Bińkowski et al. (2018); Dziugaite et al. (2015); Li et al. (2015). Optimizing the C2ST classifier at the same time as the parameters of our generative model is similar to training a GAN (Goodfellow et al., 2014), but for simplicity we instead optimized for the closed-form optimal C2ST as for this toy example we have access to true densities. While FID can be used as optimization target in principle (Mathiasen & Hvilshøj, 2021), its applicability to our toy example here is less obvious, so we excluded it here.

In order to fit the (miss-specified) Gaussian model,

$$p(x) = \mathcal{N}(\mu, \Sigma)$$

to the ground truth distribution $p_{\text{true}}$, which is a mixture of Gaussians, we proceed as follows. Let $CC^{\mathsf{T}}$ denote the Cholesky decomposition of $\Sigma$. We compute gradients with respect to $\mu$ and $C$ by using the reparameterization trick; by generating samples as $\mu + C\epsilon$, with $\epsilon \sim \mathcal{N}(0, \mathsf{I})$.

For Wasserstein we used as loss the Sliced Wasserstein distance, for MMD, we used a Gaussian Kernel with bandwidth set according the median heuristic.

For C2ST, we can evaluate the probability densities of samples from both the learned Gaussian and the ground truth mixture of Gaussians, so we minimize the accuracy of the closed-form optimal classifier. For each sample, we evaluate the log-probability density of the sample under each distribution, softmax the two resulting values, and use those as the classifier predicted probabilities. We then use binary cross-entropy as the loss function.

We used the ADAM optimizer (Kingma & Ba, 2015), with learning rate=0.01 and default momentums, using 2500 epochs of 10000 samples.

### A.8.2   Fitting a mixture of Gaussians with Expectation-Maximisation

We also include an example where the model we fit is well-specified, which in this case means it is also a mixture of two Gaussians (Fig. S3. As directly optimizing a mixture distribution with gradient descent is not straightforward, we used an Expectation-Maximization algorithm (where we use the distances instead of the log-likelihood in the maximisation step)

Our model is specified by

$$p(x) = w_1 \mathcal{N}(\mu_1, \Sigma_1) + w_2 \mathcal{N}(\mu_2, \Sigma_2)$$

which we can write as a latent-variable model, where the latent variables are the cluster assignments:

$$p(x) = \sum_{k=1}^{2} p(x|z = k)p(z = k),$$

with $p(x|z = k) = \mathcal{N}(\mu_k, \Sigma_k)$ and $p(z = k) = w_k$.

We then iteratively performed the following two steps to optimise the model.

**E-step**: For each of the $N$ datapoints $\hat{x}_i$ from $p_{\text{true}}$, we calculated the probability of it belonging to mixture component 1 or 2:

$$p(z = k|\hat{x}_i) = \frac{p(\hat{x}_i|z = k)p(z = k)}{\sum_{k'}^{2} p(\hat{x}_i|z = k')p(z = k')}.$$

**M-step**: We updated the mixture weights according to:

$$w_k = \frac{1}{N} \sum_{i}^{N} p(z = k|\hat{x}_i)$$

Next, for each of the $N$ datapoints, we first sampled a cluster assignment according to $z_i \sim p(z|\hat{x}_i)$. Then for each group of $N_k$ datapoints assigned to cluster $k$ we sampled $N_k$ times according to $x_i \sim p(x|z_i)$, again using the reparameterisation trick. As before we computed the loss using a statistical distance, now separately for the two groups of samples assigned to either mixture component, and used gradient to optimise $\mu_k, \Sigma_k$

Again, we used the ADAM optimizer (Kingma & Ba, 2015), with learning rate=0.01 and default momentums, using 2000 epochs of 5000 samples.

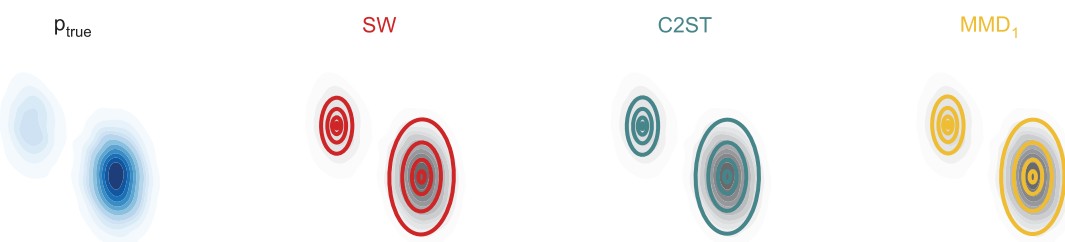

Figure S3: **Optimising distances in a well-specified model setting** When fitting a well-specified model (here, a mixture of two Gaussians), by using different distances in the loss, we can see that each model converges to the global optimum. Plotted are the contour lines of .25, .75, 1, and 2.5 standard deviations of the fitted Gaussians multiplied by their corresponding mixture weight.

### A.9 Additional scaling experiments with different sample size budgets and ranges

In Fig. 8, we evaluated the robustness of the measures against the number of samples and the dimensionality of the data. We observed notably poor performance of the measures in scenarios with limited data. Here, we further examine the performance of the distances across datasets of varying sample sizes, particularly for small sample set sizes, ranging from only 8 to 80 samples per set (Fig. S4). We examine three distinct data configurations where the distinction between the true and approximated distributions progressively increases from subpanels S4 a to c. Across all distances, it becomes evident that the larger the disparity between the two distributions, the fewer samples are needed for differentiation. In the experiment where all dimensions are mean-shifted by one, a sample size of 8 is sufficient to distinguish between the distributions. However, for less distinct distributions, such as the unimodal Gaussian or a mean-shift by one in only one dimension (Supp. Fig. S4 a, b), all distances exhibit poor performance in distinguishing between the distributions.

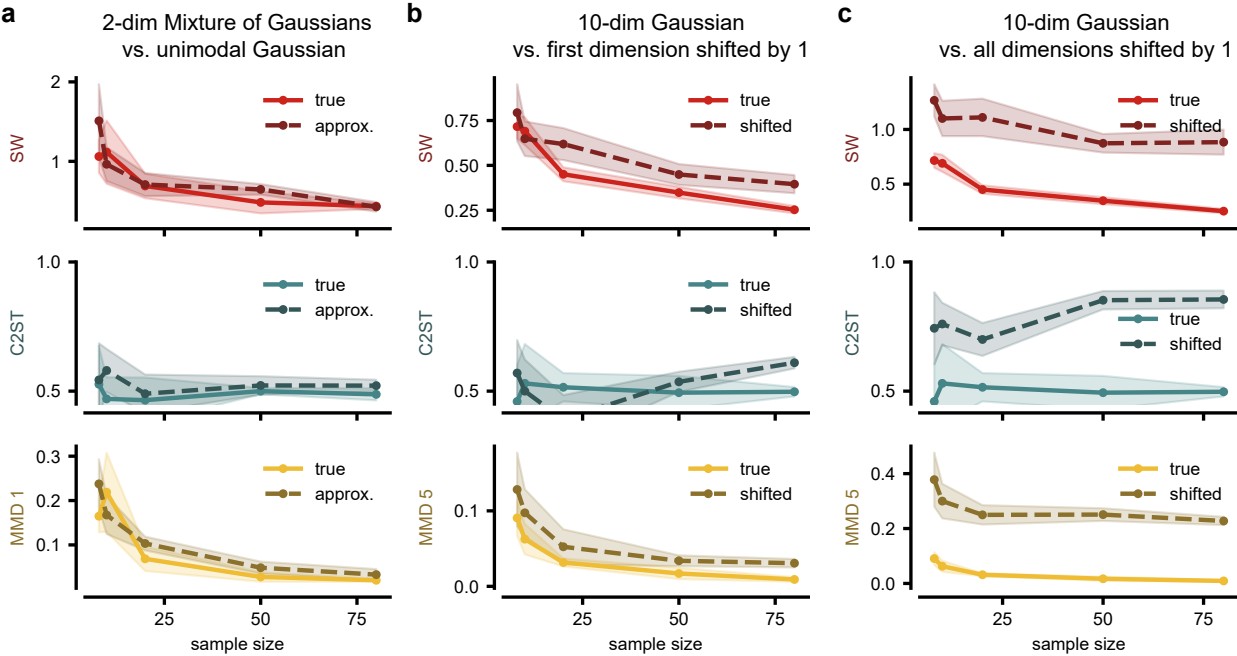

Figure S4: **The larger the difference between two distributions the fewer samples suffice to tell the true and shifted distribution apart.** We compare sample sets with varying sample sizes (between 8 and 80 samples per set) of a 'true' distribution either with a second dataset of the same distribution or with a sample set from an approximated/shifted distribution. We show the mean and standard deviation over five runs of randomly sampled data. **(a)** Distances for the 2d-MoG example shown in Fig. 1 compared to samples from a unimodal Gaussian approximation with the same mean and covariance. **(b)** Distances for a ten-dimensional standard normal distribution, for which the first dimension is shifted by one for the shifted example. **(c)** Distances for a ten-dimensional standard normal distribution, for which *all* dimensions are shifted by one for the shifted example.

## A.10  Additional scaling experiments for different dimensionality of the data

When comparing the robustness of the measures with respect to the dimensionality of the data in Fig. 8, we observed a degradation in the ability to distinguish between distributions as dimensionalities increased. Notably, only the C2ST measure retained the capability to distinguish between the two distributions in higher dimensions which is aligned with the intuition that a classifier can easily pick up on differences in a single dimension. Extending this analysis, Fig. S5 presents similar experiments conducted on datasets where we compare an n-dimensional standard normal distribution with one where either all dimensions are mean-shifted by one (thus aligning with the C2ST experiment in Fig. 5b) or where all variances are increased by one. Fig. S5a corresponds to the experiment outlined in Fig. 8c on dimensionality. The bandwidth parameter for the MMD distance has been adjusted to suit the particular data configuration and is represented by the integer in the y-axis label. Generally, we notice that the Sliced-Wasserstein distance and MMD face difficulties in higher-dimensional spaces, especially when handling distributions that are only slightly distinct if the respective hyperparameters are kept constant across dimensions. In contrast, the C2ST distance consistently demonstrates good performance across all three experiments and for all ranges of dimensions.

## A.11  Comparisons of practical compute times of different measures

Before computing such measures, in particular for scaling experiments such as the ones presented here, where measures are calculated across a large range of sample sizes $N$ and data dimensions $D$, the practical compute

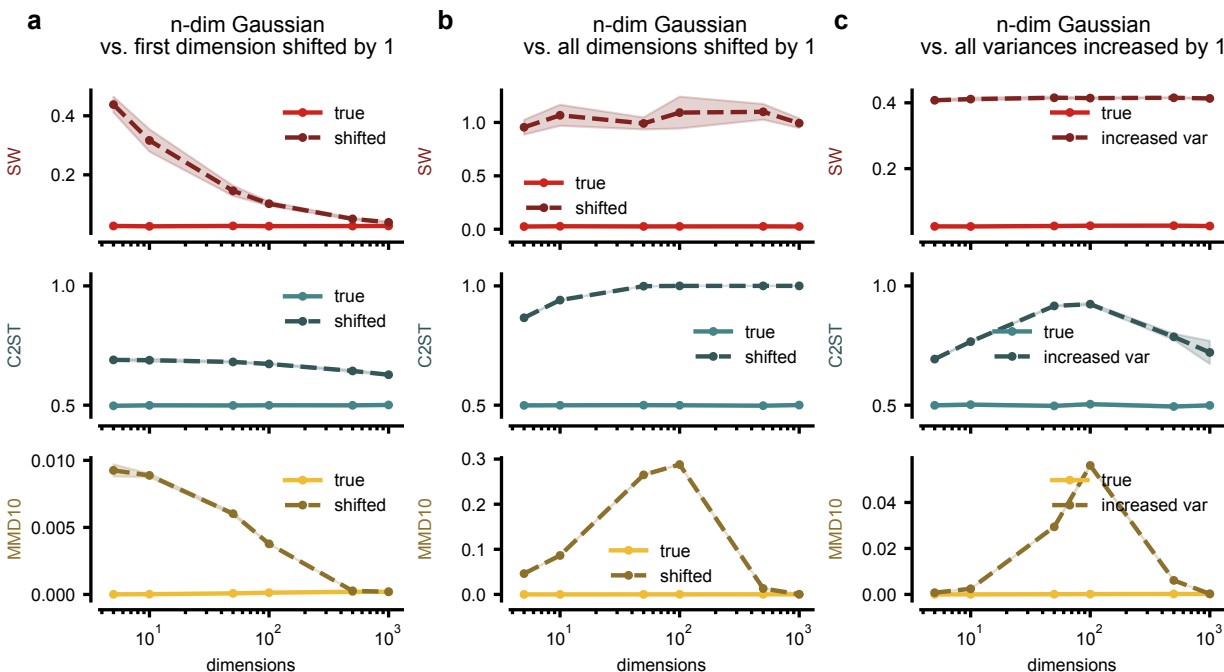

Figure S5: **The impact of dimensionality varies across distances, with certain distances facing particular challenges in higher dimensional spaces.** We compare sample sets of varying dimensionality (between 5 and 1000) of a 'true' distribution either with a second dataset of the same distribution or with a sample set from an approximated/shifted distribution. The sample size is fixed to 10k for all experiments and we show the mean and standard deviation over five runs of randomly sampled data. The bandwidth parameter in Gaussian Kernel MMD is set to 10 for all experiments. **(a)** Distances for a sample set from an n-dimensional standard normal distribution, for which *the first* dimension is shifted by one. **(b)** Distances for a sample set from an n-dimensional standard normal distribution, for which *all* dimensions are shifted by one. **(c)** Distances for a sample set from an n-dimensional standard normal distribution, for which *variances are increased* by one for all dimensions.

time of the chosen measures should be considered. Depending on the downstream application, it might be time-critical to quickly evaluate distances which might favor some measures over others. Aligned with theoretical considerations regarding sampling complexity etc. as presented in Table S2), empirical compute times vary between the different measures. Given that empirical computational times for a single measure itself vary depending on the exact implementation, compute infrastructure, and problem at hand, we list approximated compute times for running the scaling experiments in Table S3. The calculated runtime combines both the comparisons of the 'true' and the 'shifted' or 'approximated' experiments. Each experiment contains five repeats across different sampled data subsets. The sample size experiment contains eight different sample size values $N$ (50, 100, 200, 500, 1000, 2000, 3000, 4000), and the dimensionality experiment scales tests six different $D$ (5, 10, 50, 100, 500, 1000). For more details, see Section 3. The version of C2ST we use here, which is based on NN-based classifiers, takes orders of magnitude longer to compute than SW and MMD. Note, however, that alternative implementations and classifier variants could speed this up.

Table S3: CPU wallclock run times for the comparison scaling experiments in Fig.8. The runtime combines both the comparisons of the true vs. the shifted or approximated distribution. Values are rounded estimates.

| | sample size experiment | | dimensionality experiment |
|---|---|---|---|
| | 2-dim MoGs | 10-dim Gaussian | n-dim Gaussian |
| SWD | 0.3 s | 0.3 s | 1.5 s |
| C2ST | 150 s | 300 s | 1500 s |
| MMD | 3 s | 3 s | 80 s |

### A.12 Sensitivity of the MMD kernel choices and hyperparameters

The general formulation of MMD allows for a wide range of kernel choices, each potentially with their own hyperparameters. These choices can have significant effect on its behavior. We provide some experiments demonstrating of the importance of well-tuned bandwidth parameters for Gaussian Kernel MMD as well as the impact of different kernels.

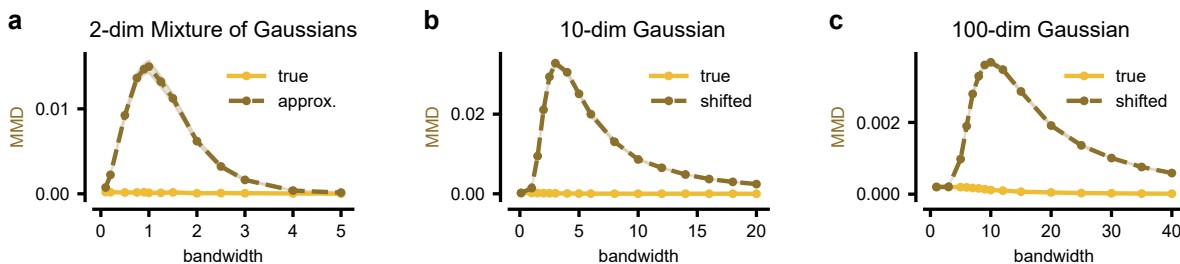

Figure S6: **The bandwidth parameter in Gaussian Kernel MMD is a sensitive parameter that requires careful selection for each dataset.** The sample size is fixed to 10k for all experiments and we show the mean and standard deviation over five runs of randomly sampled data. **(a)** $MMD^2$ distance with varying bandwidth parameters between 0.1 and 5 for the 2d-MoG example compared to samples from a unimodal Gaussian approximation with the same mean and covariance. **(b)** $MMD^2$ distance with varying bandwidth parameters between 0.1 and 20 for a 10-dimensional standard normal distribution, for which *the first* dimension is shifted by one for the shifted example. **(c)** $MMD^2$ distance with varying bandwidth parameters between 1 and 40 for a 100-dimensional standard normal distribution, for which *the first* dimension is shifted by one for the shifted example.

We first vary the bandwidth parameter with fixed sample sizes for the three example datasets used in Section 3 (Fig. S6). We show that the estimated MMD values vary significantly across bandwidths, and both setting the bandwidth too low or too high yield poor results. However, we note that the values yielded by the median heuristic (bandwidths of 1, 5, and 10 for the three datasets, respectively, as shown in the main text) are quite near the peaks of the curves at which MMD most effectively distinguishes the distributions.

We then choose a set of bandwidth parameters to compare across the scaling experiments of Section 3 (Fig. S7). Again, poor choices of bandwidth values give misleading results, but bandwidth choices guided by the median heuristic generally perform well.

Finally, we vary the kernel choice across the scaling experiments of Section 3 (Fig. S7), using both a linear kernel ($MMD_{lin}$) and the distance-induced kernel corresponding to the standard energy distance ($MMD_{en}$). As expected, the linear kernel fails to distinguish distributions with matching means (2d-MoG) but performs reasonably well for distributions with mean-offsets even at high dimensions. The energy kernel performs similarly to the Gaussian kernel, without the added dependence on sensitive hyperparameters.

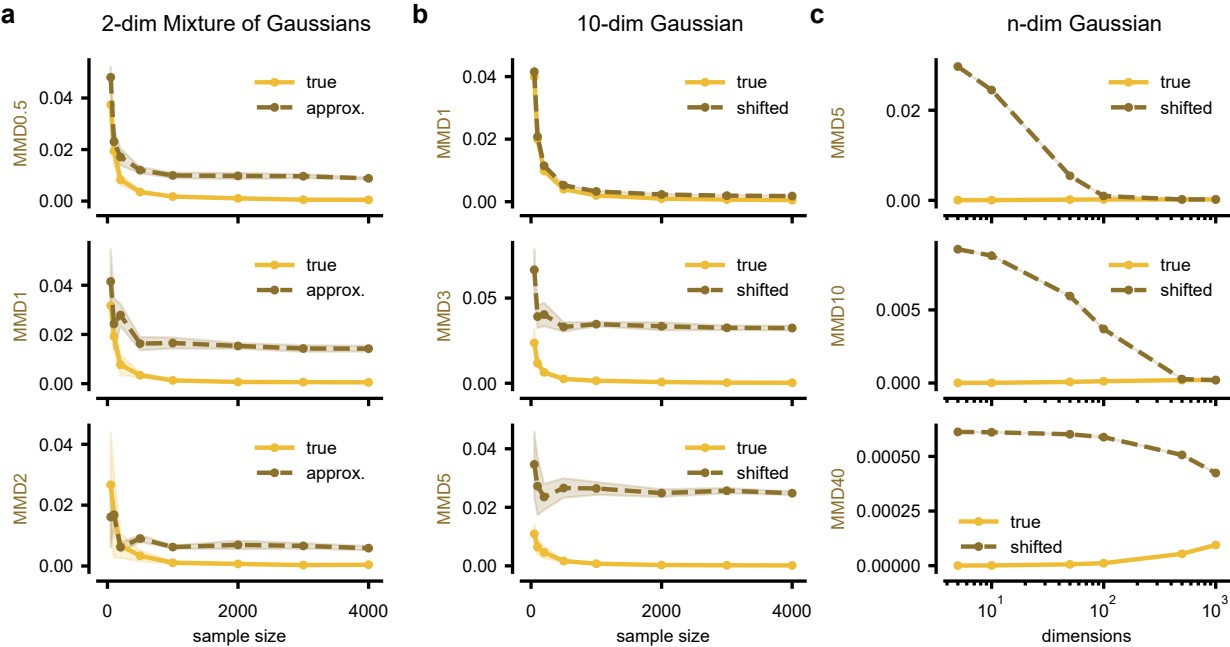

Figure S7: **Evaluating the effect of varying bandwidth parameters in Gaussian Kernel MMD for different sample sizes and dataset dimensionalities. (a,b)** We compare sample sets with varying sample sizes (between 50 and 4k samples per set) of a 'true' distribution either with a second dataset of the same distribution or with a sample set from an approximated/shifted distribution. We show the mean and standard deviation over five runs of randomly sampled data. **(a)** MMD$^2$ distance with varying bandwidth parameters (0.5, 1, 2) for the 2d-MoG example shown in Fig. 1 compared to samples from a unimodal Gaussian approximation with the same mean and covariance. **(b)** MMD$^2$ distance with varying bandwidth parameters (1, 3, 5) for a ten dimensional standard normal distribution, for which *the first* dimension is shifted by one for the shifted example. **(c)** MMD$^2$ distance with varying bandwidth parameters (5, 10, 40) based on 10k samples from a standard normal distributions with varying dimensions (between 5 and 1000). As in (b) *the first* dimension is shifted by one for the 'shifted' dataset. Here we show one run due to computational costs.

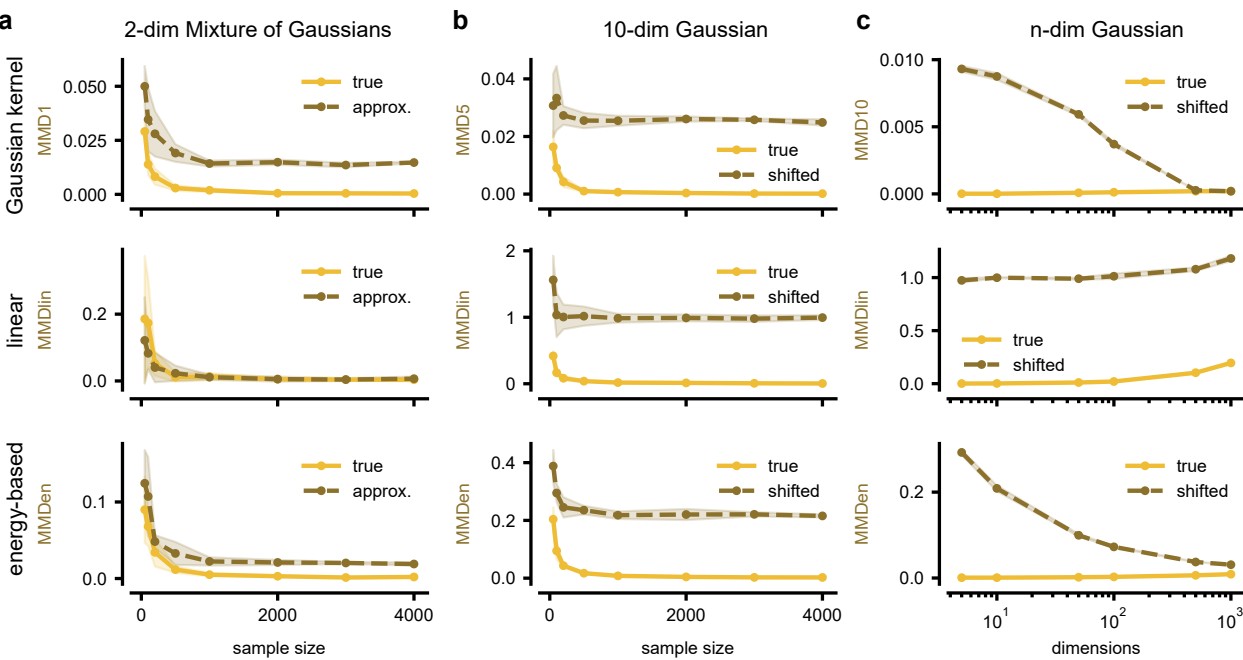

Figure S8: **Comparison of Gaussian Kernel MMD to different MMD kernels without tunable parameters.** We compare the performance of $MMD^2$ as presented in Fig.8 with a Gaussian kernel with bandwidth parameters adjusted for each dataset (1,5,10) (top row) to an MMD implementation with a linear kernel (middle row, $MMD_{lin}$; $k(x,y) = \langle x,y \rangle$) and an energy-distance based kernel, i.e., the kernel induced by the Euclidean distance Sejdinovic et al. (2013) (bottom row, $MMD_{en}$; $k(x,y) = \|x\|^p + \|x\|^p - \|x-y\|^p$, with $p = 2$). The experiments, parameters for the Gaussian kernel bandwidth (indicated in the y-labels), and sample sizes etc. are identical to Fig. 8.

### A.13 Additional results for ImageNet generative models

We generated 50,000 images using an unconditional diffusion model explicitly trained on ImageNet 64x64, as well as 100,000 class-conditional images from a conditional variant of the same model (Dockhorn et al., 2022) (we refer to it as GENIE). For additional comparison, we also generated 50,000 images using a consistency model explicitly trained on ImageNet 64x64 (Song et al., 2023) (we refer to it as CM). We compare these generated images to the ImageNet 64x64 test set.

For further comparison, we also evaluated image-generative models not specifically trained to reproduce ImageNet but designed for general-purpose image generation, such as Stable Diffusion and Midjourney (Rombach et al., 2022b; Midjourney, 2022). While these models might generate images that are more appealing to human observers, they do not necessarily produce images that align with the images contained in the ImageNet test set. We use the recently published million-scale dataset GenImage (Zhu et al., 2023). Especially, we include the models BigGAN (Brock et al., 2019), ablated diffusion model (ADM; Dhariwal & Nichol 2021), Glide (Nichol et al., 2022), Vector Quantized Diffusion Model (VQDM; Gu et al. 2022), Wukong (Wukong, 2022), Stable diffusion 1.5 ( SD1.5; Rombach et al. 2022b) and Midjourney (Midjourney, 2022).

It's important to note that not all models are specifically trained to capture ImageNet 64x64 images. For instance, models like Stable Diffusion and Midjourney are trained on much larger datasets (Schuhmann et al., 2022; Lin et al., 2014). Additionally, most of the models mentioned, except for ADM and BigGAN, are text-to-image generative models. To minimize significant distribution shifts, these models were prompted with the phrase "photo of [ImageNet class]" Zhu et al. (2023). Furthermore, all the other models generate images of larger resolution, which we resized to 64x64. Hence, we also only compare low-resolution features of natural images.

We evaluate each of the metrics on three random subsets, each consisting of 40,000 image embeddings. We show the average value for each model and metric in Table S4. Interestingly, both what is considered "closest" to ImageNet, as well as the relative ranking differs for different metrics, although with some consistent trends. Overall, the most recent unconditional models trained on ImageNet 64x64 perform best, as expected. However, which one is considered best differs for different metrics. Metrics that consider similar features of the distribution i.e., FID and $\text{MMD}_{\text{poly}}$ (statistics up to order 2 or 3) prefer GENIE, whereas universally consistent metrics do prefer CM (SWD and $\text{MMD}_{64}$). The estimated C2ST values differ based on the chosen classifier.

Overall, this analysis highlights that the choice of metric (i.e., what features it compares) and the specific implementation details (such as the classifier in C2ST estimates) matter and can lead to varying results.

Table S4: **Evaluating discrepancy to the ImageNet test set.** Each row presents various metrics computed on the Inception v3 embeddings of images. Columns correspond to different generative models. The initial three models are trained on ImageNet 64x64, serving as the reference point for comparison. Subsequent models are trained on alternative datasets or higher-resolution versions. In bold, we highlight the lowest value for each section of the table.

| | GENIE | CM | BigGAN | ADM | Glide | VQDM | WK | SD1.5 | Midjourney |
|---|---|---|---|---|---|---|---|---|---|
| FID | $\mathbf{5.1 \cdot 10^0}$ | $5.3 \cdot 10^0$ | $1.2 \cdot 10^1$ | $1.1 \cdot 10^1$ | $1.1 \cdot 10^1$ | $\mathbf{9.8 \cdot 10^0}$ | $1.1 \cdot 10^1$ | $1.2 \cdot 10^1$ | $1.0 \cdot 10^1$ |
| SWD | $2.3 \cdot 10^{-2}$ | $\mathbf{2.2 \cdot 10^{-2}}$ | $5.3 \cdot 10^{-2}$ | $5.1 \cdot 10^{-2}$ | $4.9 \cdot 10^{-2}$ | $\mathbf{4.0 \cdot 10^{-2}}$ | $4.8 \cdot 10^{-2}$ | $5.0 \cdot 10^{-2}$ | $4.5 \cdot 10^{-2}$ |
| $\text{MMD}_{64}$ | $7.0 \cdot 10^{-5}$ | $\mathbf{6.3 \cdot 10^{-5}}$ | $1.9 \cdot 10^{-4}$ | $1.9 \cdot 10^{-4}$ | $1.8 \cdot 10^{-4}$ | $\mathbf{1.5 \cdot 10^{-4}}$ | $1.9 \cdot 10^{-4}$ | $1.9 \cdot 10^{-4}$ | $1.7 \cdot 10^{-4}$ |
| $\text{MMD}_{\text{lin}}$ | $2.5 \cdot 10^{-1}$ | $\mathbf{2.3 \cdot 10^{-1}}$ | $6.3 \cdot 10^{-1}$ | $6.3 \cdot 10^{-1}$ | $6.2 \cdot 10^{-1}$ | $\mathbf{5.2 \cdot 10^{-1}}$ | $6.2 \cdot 10^{-1}$ | $6.4 \cdot 10^{-1}$ | $6.1 \cdot 10^{-1}$ |
| $\text{MMD}_{\text{poly}}$ | $\mathbf{1.1 \cdot 10^4}$ | $1.6 \cdot 10^4$ | $3.1 \cdot 10^4$ | $3.4 \cdot 10^4$ | $3.2 \cdot 10^4$ | $\mathbf{2.2 \cdot 10^4}$ | $3.7 \cdot 10^4$ | $3.1 \cdot 10^4$ | $2.9 \cdot 10^4$ |
| $\text{C2ST}_{\text{knn}}$ | $0.70$ | $\mathbf{0.69}$ | $\mathbf{0.81}$ | $0.82$ | $0.82$ | $0.82$ | $0.83$ | $0.83$ | $0.82$ |
| $\text{C2ST}_{\text{nn}}$ | $\mathbf{0.72}$ | $0.77$ | $0.87$ | $0.85$ | $0.85$ | $\mathbf{0.85}$ | $0.86$ | $0.86$ | $0.95$ |

### A.14    Details about scientific application examples

For the motion discrimination task, we used the decision times of a single animal during both correct and erroneous trials with dot motion coherence of 12.8%, leading to a one-dimensional dataset of 1023 samples. From these 80% were used as a train set and 20% as a test set DDMs were implemented using the *pyDDM* toolbox (Shinn et al., 2020). DDM1 used a linear drift and exponential decision boundaries. In contrast, DDM2 used a constant drift and a constant decision boundary. Both were sampled 1,000 times to create the two synthetic datasets. The real chest X-ray dataset consists of 70,153 train samples and 25,596 test samples, the generated datasets from PGGAN and SD consist of 10,000 and 2,352 samples respectively.

In both applications we computed metrics between pairs of 10 random subsets from the compared distributions (scatter points on the violin plots). We computed the MMD with a bandwidth of 50 for the medical imaging datasets and a bandwidth of 0.5 for the decision time dataset.

## References for Table S1

[1] J. H. Macke, L. Buesing, J. P. Cunningham, B. M. Yu, K. V. Shenoy, and M. Sahani. Empirical models of spiking in neural populations. In *Advances in Neural Information Processing Systems*, 2011.

[2] S. Linderman, M. Johnson, A. Miller, R. Adams, D. Blei, and L. Paninski. Bayesian learning and inference in recurrent switching linear dynamical systems. In *Proceedings of the 20th International Conference on Artificial Intelligence and Statistics*, 2017.

[3] C. Pandarinath, D. J. O'Shea, J. Collins, R. Jozefowicz, S. D. Stavisky, J. C. Kao, E. M. Trautmann, M. T. Kaufman, S. I. Ryu, L. R. Hochberg, J. M. Henderson, K. V. Shenoy, L. F. Abbott, and D. Sussillo. Inferring single-trial neural population dynamics using sequential auto-encoders. *Nature Methods*, 2018.

[4] T. D. Kim, T. Z. Luo, T. Can, K. Krishnamurthy, J. W. Pillow, and C. D. Brody. Flow-field inference from neural data using deep recurrent networks. *bioRxiv*, 2023.

[5] M. Brenner, F. Hess, G. Koppe, and D. Durstewitz. Integrating multimodal data for joint generative modeling of complex dynamics. In *Forty-first International Conference on Machine Learning*, 2024.

[6] M. Molano-Mazon, A. Onken, E. Piasini, and S. Panzeri. Synthesizing realistic neural population activity patterns using generative adversarial networks. *International Conference on Learning Representations*, 2018.

[7] P. Ramesh, M. Atayi, and J. H. Macke. Adversarial training of neural encoding models on population spike trains. *Real Neurons & Hidden Units: Future directions at the intersection of neuroscience and artificial intelligence @ NeurIPS 2019*, 2019.

[8] M. Bashiri, E. Walker, K.-K. Lurz, A. Jagadish, T. Muhammad, Z. Ding, Z. Ding, A. Tolias, and F. Sinz. A flow-based latent state generative model of neural population responses to natural images. In *Advances in Neural Information Processing Systems*, 2021.

[9] A. Schulz, J. Vetter, R. Gao, D. Morales, V. Lobato-Rios, P. Ramdya, P. J. Gonçalves, and J. H. Macke. Modeling conditional distributions of neural and behavioral data with masked variational autoencoders. *bioRxiv*, 2024.

[10] M. Pals, A. E. Sağtekin, F. Pei, M. Gloeckler, and J. H. Macke. Inferring stochastic low-rank recurrent neural networks from neural data. *arXiv preprint arXiv:2406.16749*, 2024.

[11] J. Kapoor, A. Schulz, J. Vetter, F. Pei, R. Gao, and J. H. Macke. Latent diffusion for neural spiking data. *arXiv preprint arXiv:2407.08751*, 2024.

[12] J. D. McCart, A. R. Sedler, C. Versteeg, D. Mifsud, M. Rigotti-Thompson, and C. Pandarinath. Diffusion-based generation of neural activity from disentangled latent codes. *arXiv preprint arXiv:2407.21195*, 2024.

[13] M. Dowling, Y. Zhao, and I. M. Park. Large-scale variational gaussian state-space models. *arXiv preprint arXiv:2403.01371*, 2024.

[14] R. Lopez, J. Regier, M. B. Cole, M. I. Jordan, and N. Yosef. Deep generative modeling for single-cell transcriptomics. *Nature methods*, 2018.

[15] M. Marouf, P. Machart, V. Bansal, C. Kilian, D. S. Magruder, C. F. Krebs, and S. Bonn. Realistic in silico generation and augmentation of single-cell rna-seq data using generative adversarial networks. *Nature communications*, 2020.

[16] S. Lall, S. Ray, and S. Bandyopadhyay. LSH-GAN enables in-silico generation of cells for small sample high dimensional scRNA-seq data. *Communications Biology*, 2022.

[17] Y. Zinati, A. Takiddeen, and A. Emad. Groundgan: Grn-guided simulation of single-cell rna-seq data using causal generative adversarial networks. *Nature Communications*, 2024.

[18] W. Tang, R. Liu, H. Wen, X. Dai, J. Ding, H. Li, W. Fan, Y. Xie, and J. Tang. A general single-cell analysis framework via conditional diffusion generative models. *bioRxiv*, 2023.

[19] R. Cannoodt, W. Saelens, L. Deconinck, and Y. Saeys. Spearheading future omics analyses using dyngen, a multi-modal simulator of single cells. *Nature Communications*, 2021.

[20] O. Lindenbaum, J. Stanley, G. Wolf, and S. Krishnaswamy. Geometry based data generation. *Advances in Neural Information Processing Systems*, 2018.

[21] L. Zappia, B. Phipson, and A. Oshlack. Splatter: simulation of single-cell RNA sequencing data. *Genome biology*, 2017.

[22] H. L. Crowell, C. Soneson, P.-L. Germain, D. Calini, L. Collin, C. Raposo, D. Malhotra, and M. D. Robinson. Muscat detects subpopulation-specific state transitions from multi-sample multi-condition single-cell transcriptomics data. *Nature communications*, 2020.

[23] W. V. Li and J. J. Li. A statistical simulator scdesign for rational scRNA-seq experimental design. *Bioinformatics*, 2019.

[24] T. Sun, D. Song, W. V. Li, and J. J. Li. scDesign2: a transparent simulator that generates high-fidelity single-cell gene expression count data with gene correlations captured. *Genome biology*, 2021.

[25] N. R. Stillman and R. Mayor. Generative models of morphogenesis in developmental biology. In *Seminars in Cell & Developmental Biology*, 2023.

[26] D. J. Waibel, E. Röell, B. Rieck, R. Giryes, and C. Marr. A diffusion model predicts 3d shapes from 2d microscopy images. In *2023 IEEE 20th International Symposium on Biomedical Imaging (ISBI)*, 2023.

[27] A. Zaritsky, A. R. Jamieson, E. S. Welf, A. Nevarez, J. Cillay, U. Eskiocak, B. L. Cantarel, and G. Danuser. Interpretable deep learning uncovers cellular properties in label-free live cell images that are predictive of highly metastatic melanoma. *Cell systems*, 2021.

[28] C. J. Soelistyo, G. Vallardi, G. Charras, and A. R. Lowe. Learning biophysical determinants of cell fate with deep neural networks. *Nature Machine Intelligence*, 2022.

[29] R. L. Satcher and C. F. Dewey. Theoretical estimates of mechanical properties of the endothelial cell cytoskeleton. *Biophysical journal*, 1996.

[30] D. Stamenović, J. J. Fredberg, N. Wang, J. P. Butler, and D. E. Ingber. A microstructural approach to cytoskeletal mechanics based on tensegrity. *Journal of Theoretical Biology*, 1996.

[31] D. Stamenović and D. E. Ingber. Models of cytoskeletal mechanics of adherent cells. *Biomechanics and modeling in mechanobiology*, 2002.

[32] M. F. Coughlin and D. Stamenović. A prestressed cable network model of the adherent cell cytoskeleton. *Biophysical journal*, 2003.

[33] G. Jouvet, G. Cordonnier, B. Kim, M. Lüthi, A. Vieli, and A. Aschwanden. Deep learning speeds up ice flow modelling by several orders of magnitude. *Journal of Glaciology*, 2022.

[34] G. Jouvet. Inversion of a stokes glacier flow model emulated by deep learning. *Journal of Glaciology*, 2023.

[35] G. Jouvet and G. Cordonnier. Ice-flow model emulator based on physics-informed deep learning. *Journal of Glaciology*, 2023.

[36] J. Bolibar, F. Sapienza, F. Maussion, R. Lguensat, B. Wouters, and F. Pérez. Universal differential equations for glacier ice flow modelling. *Geoscientific Model Development*, 2023.

[37] V. Verjans and A. Robel. Accelerating subglacial hydrology for ice sheet models with deep learning methods. *Geophysical Research Letters*, 2024.

[38] R. Winkelmann, M. A. Martin, M. Haseloff, T. Albrecht, E. Bueler, C. Khroulev, and A. Levermann. The potsdam parallel ice sheet model (PISM-PIK) – Part 1: Model description. *The Cryosphere*, 2011.

[39] E. Larour, H. Seroussi, M. Morlighem, and E. Rignot. Continental scale, high order, high spatial resolution, ice sheet modeling using the ice sheet system model (ISSM). *Journal of Geophysical Research: Earth Surface*, 2012.

[40] O. Gagliardini, T. Zwinger, F. Gillet-Chaulet, G. Durand, L. Favier, B. de Fleurian, R. Greve, M. Malinen, C. Martín, P. Råback, J. Ruokolainen, M. Sacchettini, M. Schäfer, H. Seddik, and J. Thies. Capabilities and performance of elmer/ice, a new-generation ice sheet model. *Geoscientific Model Development*, 2013.

[41] R. Lam, A. Sanchez-Gonzalez, M. Willson, P. Wirnsberger, M. Fortunato, F. Alet, S. Ravuri, T. Ewalds, Z. Eaton-Rosen, W. Hu, A. Merose, S. Hoyer, G. Holland, O. Vinyals, J. Stott, A. Pritzel, S. Mohamed, and P. Battaglia. Learning skillful medium-range global weather forecasting. *Science*, 2023.

[42] T. R. Shaham, T. Dekel, and T. Michaeli. SinGAN: Learning a generative model from a single natural image. In *2019 IEEE/CVF International Conference on Computer Vision (ICCV)*, 2019.

[43] D. J. Gagne II, S. E. Haupt, W. D. Nychka, and G. Thompson. Interpretable deep learning for spatial analysis of severe hailstorms. *Monthly Weather Review*, 2019.

[44] J. Côté, S. Gravel, A. Méthot, A. Patoine, M. Roch, and A. Staniforth. The operational CMC–MRB global environmental multiscale (GEM) model. Part i: Design considerations and formulation. *Monthly Weather Review*, 1998.

[45] L. Zhou, S.-J. Lin, J.-H. Chen, L. M. Harris, X. Chen, and S. L. Rees. Toward convective-scale prediction within the next generation global prediction system. *Bulletin of the American Meteorological Society*, 2019.

[46] L. Magnusson, J.-R. Bidlot, M. Bonavita, A. R. Brown, P. A. Browne, G. D. Chiara, M. Dahoui, S. T. K. Lang, T. McNally, K. S. Mogensen, F. Pappenberger, F. Prates, F. Rabier, D. S. Richardson, F. Vitart, and S. Malardel. ECMWF activities for improved hurricane forecasts. *Bulletin of the American Meteorological Society*, 2019.

[47] D. M. Anstine and O. Isayev. Generative models as an emerging paradigm in the chemical sciences. *Journal of the American Chemical Society*, 2023.

[48] R. Gómez-Bombarelli, J. N. Wei, D. Duvenaud, J. M. Hernández-Lobato, B. Sánchez-Lengeling, D. Sheberla, J. Aguilera-Iparraguirre, T. D. Hirzel, R. P. Adams, and A. Aspuru-Guzik. Automatic chemical design using a data-driven continuous representation of molecules. *ACS central science*, 2018.

[49] R. Winter, F. Montanari, A. Steffen, H. Briem, F. Noé, and D.-A. Clevert. Efficient multi-objective molecular optimization in a continuous latent space. *Chemical science*, 2019.

[50] J. Lim, S. Ryu, J. W. Kim, and W. Y. Kim. Molecular generative model based on conditional variational autoencoder for de novo molecular design. *Journal of cheminformatics*, 2018.

[51] B. Sanchez-Lengeling, C. Outeiral, G. L. Guimaraes, and A. Aspuru-Guzik. Optimizing distributions over molecular space. an objective-reinforced generative adversarial network for inverse-design chemistry (ORGANIC). *ChemRxiv*, 2017.

[52] N. De Cao and T. Kipf. MolGAN: An implicit generative model for small molecular graphs. *ICML 2018 workshop on Theoretical Foundations and Applications of Deep Generative Models*, 2018.

[53] E. Hoogeboom, V. G. Satorras, C. Vignac, and M. Welling. Equivariant diffusion for molecule generation in 3d. In *International conference on machine learning*, 2022.

[54] L. Huang, H. Zhang, T. Xu, and K.-C. Wong. Mdm: Molecular diffusion model for 3d molecule generation. In *Proceedings of the AAAI Conference on Artificial Intelligence*, 2023.

[55] L. Wu, C. Gong, X. Liu, M. Ye, and Q. Liu. Diffusion-based molecule generation with informative prior bridges. *Advances in Neural Information Processing Systems*, 2022.

[56] M. Popova, O. Isayev, and A. Tropsha. Deep reinforcement learning for de novo drug design. *Science advances*, 2018.

[57] N. Ståhl, G. Falkman, A. Karlsson, G. Mathiason, and J. Bostrom. Deep reinforcement learning for multiparameter optimization in de novo drug design. *Journal of chemical information and modeling*, 2019.

[58] S. K. Gottipati, B. Sattarov, S. Niu, Y. Pathak, H. Wei, S. Liu, S. Blackburn, K. Thomas, C. Coley, J. Tang, et al. Learning to navigate the synthetically accessible chemical space using reinforcement learning. In *International conference on machine learning*, 2020.

[59] J. Regier, A. Miller, J. McAuliffe, R. Adams, M. Hoffman, D. Lang, D. Schlegel, and M. Prabhat. Celeste: Variational inference for a generative model of astronomical images. In *International Conference on Machine Learning*, 2015.

[60] F. Lanusse, R. Mandelbaum, S. Ravanbakhsh, C.-L. Li, P. Freeman, and B. Póczos. Deep generative models for galaxy image simulations. *Monthly Notices of the Royal Astronomical Society*, 2021.

[61] M. J. Smith and J. E. Geach. Generative deep fields: arbitrarily sized, random synthetic astronomical images through deep learning. *Monthly Notices of the Royal Astronomical Society*, 2019.

[62] R. Mandelbaum, C. M. Hirata, A. Leauthaud, R. J. Massey, and J. Rhodes. Precision simulation of ground-based lensing data using observations from space. *Monthly Notices of the Royal Astronomical Society*, 2012.

[63] B. T. Rowe, M. Jarvis, R. Mandelbaum, G. M. Bernstein, J. Bosch, M. Simet, J. E. Meyers, T. Kacprzak, R. Nakajima, J. Zuntz, et al. GALSIM: the modular galaxy image simulation toolkit. *Astronomy and Computing*, 2015.

