# OpenReview forum: "A Practical Guide to Sample-based Statistical Distances for Evaluating Generative Models in Science"
_TMLR — Accepted by TMLR_

### Review · Reviewer_Yh3B · 2024-04-28

**Summary Of Contributions:**

The work provides an overview, some intuitive and experimental comparisons of measures of dissimilarity which are typically used to assess the quality of generative models in practice.

**Audience:**

Yes

**Claims And Evidence:**

Yes

**Requested Changes:**

I think the authors should fix most of the issues to which I pointed above. Overall, I am quite confident that the current survey/guide is not comprehensive enough and misses a lot of crucial details and aspects about the evaluation metric for generative models.

I would like to have a clear understanding how the current survey is better than any other existing surveys/related papers. What are the closely related surveys/guides? What is the novelty/improvement compared to them that will motivate the practitioner to read/use your survey rather than the existing ones? For example, a randomly googled recent related survey [3] already looks much more complete than the current paper (although it seems to be focused at slightly different aspects).

Lastly, the community also actively works on developing novel metrics/measures of dissimilarity to comapre probability distributions. It may be beneficial for the current survey/guide to at least mention them. For example, paper [3] proposes an alternatie to FID based on central kernel alignment (CKA). Paper [4] proposes some topology-based metrics, etc. More broadly, I think it would be beneficial for the current paper to at least mention such and related developments in the field to tell the practicioner some other directions to look at which may be particularly useful for his type of data and problem.

[3] Yang, C., Zhang, Y., Bai, Q., Shen, Y., & Dai, B. (2023). Revisiting the evaluation of image synthesis with gans. Advances in Neural Information Processing Systems, 36, 9518-9542..

[4] Barannikov, S., Trofimov, I., Sotnikov, G., Trimbach, E., Korotin, A., Filippov, A., & Burnaev, E. (2021). Manifold Topology Divergence: a Framework for Comparing Data Manifolds. Advances in neural information processing systems, 34, 7294-7305.

**Strengths And Weaknesses:**

**Strength:**
- The work overviews several popular metrics, including Freschet distance, classifier-based score and MMD-type metrics, as well as particular ways to use them (with/without the embedding space).
- Experimental comparison of their scalability (w.r.t. dimensionality, number of samples, etc.) is present in several toy and real-world setups and some conclusions (recommendations to use this or that metric) are made based on it.

**Weaknesses.**
Unfortunately, I think that the presented study is not comprehensive enough and lacks a tremendous amount of important things that should be (in my opinion) included in such a survey. I will list such things below together with the weaknesses.

- The paper pays practical attention to sample properties of the metrics (i.e., how many samples do you need to accurately estimate the metric), but barely mentions theoretical properties, namely, sample complexity. I think it is quite important to mention this aspect. What is the sample complexity? Which metrics have biased estimators and which have unbiased? Biased metrics a priory are expected to perform worse and be less reliable.
- It would be good to have a single table summarizing the key theoretical and practical properties, advantages and limitations of the metrics studied. For example, the above-mentioned theoretical sample complexity, the theoretical computational complexity for N samples in dimension D, practical computational time, etc.
- Studying the SW distance as a metric seems a little bit weird to me and rather far-fetched. Indeed, (as the authors say themself) “currently not extensively used in literature for evaluation”. Are there any existing papers which use it for evaluation (not for GAN training?). Also, I miss a clear discussion/evaluation related to the amount of required projections to accurately estimate SW.
- The study of MMD is too quick in the sense that the authors just use a single kernel with a fixed (chosen according to some rule) bandwidth (+there is a small study in appendix of different width). I would expect a more thorough study of the effect of different kernels beyond the considered one. For example, there are kernels for which you do not need to tune the hyper-parameters such as the bandwidth, e.g., the energy-based kernels [1] defining the standard energy distance. How do such kernels perform?
- One of the essential aspects of the generative modeling is to cover the entire distribution, i.e., not to drop/collapse any models of the distribution. In fact, this is the important part of the well-known generative modeling trilemma [2]. In the current paper, I did not find any convincing discussion of how the metrics take this aspect into account. (the word “collapse” appears only once in the paper). In fact, the community has invented several ways to address these aspects, e.g., the recall metric, but the current paper completely skips this and does not even mention.
- The paper works with MMD and FID, but it seems like (probably I miss something) it does not directly mention the well-celebrated Kernel Inception distance, an MMD-based analog of FID (although the corresponding paper is cited).
- For MMD, the authors give its clear formal definition in terms of distributions and then explain how to estimate it using the samples (provide an unbiased estimator). For the other metrics, the authors just directly write how to estimate them from samples or even explain this in words. This is not very good, especially in the case of Wasserstein, where the sample-based estimator is tremendously biased compared to the true Wasserstein between the underlying distributions. I think this might be misleading for the readers who are not familiar with such things and can lead to confusion. I recommend the authors the everywhere clearly explain the metric theoretically, and separately discuss the practical estimation aspects, just as they did in MMD.

[1] Sejdinovic, D., Sriperumbudur, B., Gretton, A., & Fukumizu, K. (2013). Equivalence of distance-based and RKHS-based statistics in hypothesis testing. The annals of statistics, 2263-2291.

[2] Xiao, Z., Kreis, K., & Vahdat, A. (2021, October). Tackling the Generative Learning Trilemma with Denoising Diffusion GANs. In International Conference on Learning Representations.

**Other:**
- What does this phrase mean? “However, a disadvantage of the Wasserstein distance is its transparency: The numerical value of the Wasserstein distance has no intuitive interpretation due to its definition in terms of optimal transport maps”
- Do I understand correctly, that in all the cases (scientific applications) you compute the metric of generated images w.r.t. the test (hold-out set of data) which has been used for training? Can you also add the metric of test vs. train data in all the cases for comparison (Figs. 10., 11)?

---

> ### Author Response · Authors · 2024-06-10
> **Rebutal Yh3B**
>
> We thank the reviewer for the insightful and extensive feedback on our manuscript. The reviewer recognizes that our work “overviews several popular metrics” and provides an “experimental comparison of their scalability” and some “recommendations to use” these. We appreciate the reviewer's time in providing a comprehensive review, highlighting several areas to improve our manuscript.
>
> We have carefully considered each of your points and have already made several changes to address them. Below, we summarize the changes  we've implemented in the revision:
>
> > “The paper pays practical attention to sample properties of the metrics (i.e., how many samples do you need to accurately estimate the metric), but barely mentions theoretical properties …” “
>
> >It would be good to have a single table summarizing the key theoretical and practical properties, advantages and limitations of the metrics studied.”
>
> We appreciate the reviewer's feedback. Our paper intentionally focuses on the practical side of these metrics rather than diving deep into theory.  However, we acknowledge the importance of discussing some key theoretical aspects, such as sample complexity and bias in sample-based estimators. We have added a discussion of this for the metrics where this property was missing (e.g., lines 147-151) and additionally summarized in Table 1 some key properties of the metrics.
> Given that the practical computational time varies significantly depending on the exact implementation, compute infrastructure and problem at hand, we decided to include information about practical compute time in the supplements. The table lists the compute time for each of the scaling experiments presented in Figure 8.
>
> > Studying the SW distance as a metric seems a little bit weird to me and rather far-fetched. Indeed, (as the authors say themself) “currently not extensively used in literature for evaluation”. Are there any existing papers which use it for evaluation (not for GAN training?). Also, I miss a clear discussion/evaluation related to the amount of required projections to accurately estimate SW.
>
> We partially agree with the reviewer on this comment. Slicing-based distance have started to gain traction relatively recently. As such, there have not been many traditional uses of sliced metrics in the context of generative models. Recent works, however, used the Sliced Wasserstein distance to train not only GANs [1,2], but also autoencoders [2], normalizing flows [3], as well as for evaluation in specific contexts [5,6].
> Generally, we also want to convey the general concept of slicing, which is useful not only for Wasserstein but also as a general technique to compute various metrics efficiently for high dimensional distributions. In our work, we find it to work well as a “general purpose” evaluation metric as it does not rely on any hyperparameters that can dramatically change the result (as it can e.g., for MMD, C2ST). We've included a discussion regarding the necessary number of projections. (Section A4, Fig. S1).
>
> > The study of MMD is too quick in the sense that the authors just use a single kernel with a fixed (chosen according to some rule) bandwidth (+there is a small study in appendix of different width). I would expect a more thorough study of the effect of different kernels beyond the considered one. For example, there are kernels for which you do not need to tune the hyper-parameters such as the bandwidth, e.g., the energy-based kernels [1] defining the standard energy distance. How do such kernels perform?
>
> Indeed, the implementation of the RBF kernel brings the inconvenience of having to adjust a bandwidth parameter, and as aptly pointed out by the reviewer, alternative kernels have not been explored sufficiently but might be particularly valuable to practitioners. To this end, we have now performed additional experiments to assess the performance of such kernels, such as a linear kernel and the proposed energy-based kernel (the kernel induced by the Euclidean distance). This proposed kernel does indeed perform comparatively well. The linear kernel fails to tell the 2MoGs apart where means are matched but is capable of distinguishing the shifted version successfully across a large sample size range as well as the dimensionality of the Gaussian - even more so than the energy-based and RBF kernels. Since the RBF kernel is still one of the most common choices for MMD, we will keep the RBF implementation in the main results and have added an additional supplementary figure (Fig. S6, S7) with the kernel comparisons. We additionally mention the link between MMD and energy distance in the main text (lines 285-289).

---

> > ### Author Response · Authors · 2024-06-10
> > **Rebutal Yh3B part II**
> >
> > > One of the essential aspects of the generative modeling is to cover the entire distribution, i.e., not to drop/collapse any models of the distribution. In fact, this is the important part of the well-known generative modeling trilemma [2]. In the current paper, I did not find any convincing discussion of how the metrics take this aspect into account. (the word “collapse” appears only once in the paper). In fact, the community has invented several ways to address these aspects, e.g., the recall metric, but the current paper completely skips this and does not even mention.
> >
> > We agree with the reviewer that this point falls short in our manuscript (although we already include several results on this topic  Fig 9c,Fig. S2, and Fig. S3). We extended the discussion of this topic in the manuscript and referred to the literature the reviewer suggested (new Section 3.5).
> >
> > > The paper works with MMD and FID, but it seems like (probably I miss something) it does not directly mention the well-celebrated Kernel Inception distance, an MMD-based analog of FID (although the corresponding paper is cited).
> >
> > Thank you for bringing this to our attention. We indeed use Kernel Inception Distances, but we did not introduce the term properly in the manuscript. In the revision, we have added an additional explanation about the relation between Kernel Inception Distance (KID)  and FID (lines 330-336, Section 2.4). Yet, as we use all metrics (SW, C2ST, and MMD) on the embeddings (Inception v3 or CLIP), we keep referring to it by whatever metric we use on the embeddings.
> >
> > > For MMD, the authors give its clear formal definition in terms of distributions and then explain how to estimate it using the samples (provide an unbiased estimator). For the other metrics, the authors just directly write how to estimate them from samples or even explain this in words. This is not very good, especially in the case of Wasserstein, where the sample-based estimator is tremendously biased compared to the true Wasserstein between the underlying distributions. I think this might be misleading for the readers who are not familiar with such things and can lead to confusion. I recommend the authors the everywhere clearly explain the metric theoretically, and separately discuss the practical estimation aspects, just as they did in MMD.
> >
> > We thank the reviewer for noticing this mismatch in how the different metrics were described. For the Wasserstein distance, we did include some details on the formal (continuous) definition in the appendix. In the revised text, we move this discussion to the main text and expand on the formal definition of the Sliced Wasserstein distance in the appendix (Section A3). Additionally, the bias of the sample-based Wasserstein distance is clearly mentioned and referenced in the main text (lines 147-151).
> >
> > > I would like to have a clear understanding how the current survey is better than any other existing surveys/related papers. What are the closely related surveys/guides? What is the novelty/improvement compared to them that will motivate the practitioner to read/use your survey rather than the existing ones? For example, a randomly googled recent related survey [3] already looks much more complete than the current paper (although it seems to be focused at slightly different aspects).
> >
> > The main value of this guide is providing an accessible entry point to understanding commonly used sample-based statistical distances. Indeed, the general nature of this article means it can neither be completely comprehensive or provide a clear-cut answer to which distance one should use. There are simply too many distances in use to cover all --- the ideal distance (or distances) one needs will depend on the specific use case. Rather, by going through four different classes of distances in detail, as well as systematically comparing them, we aim to provide a solid foundation for navigating the extensive literature on statistical distances. This indeed means that there exist other surveys that are more comprehensive when it comes to a specific use-case (e.g., GANs as the one you mentioned). We now more explicitly refer to surveys on specific topics in the introduction, and more concretely motivate what is the main value of our guide.

---

> > > ### Author Response · Authors · 2024-06-10
> > > **Rebutal Yh3B part III**
> > >
> > > > What does this phrase mean? “However, a disadvantage of the Wasserstein distance is its transparency: The numerical value of the Wasserstein distance has no intuitive interpretation due to its definition in terms of optimal transport maps”
> > >
> > > We thank the reviewer for the comment. We have attempted to clarify our meaning here by rephrasing this sentence in the manuscript: “However, a disadvantage of the Wasserstein distance is that its numerical value has no intuitive interpretation. Therefore, it is typically used to compare whether some distances are larger or smaller than others, instead of making qualitative statements about two distributions.”
> > >
> > > > Do I understand correctly, that in all the cases (scientific applications) you compute the metric of generated images w.r.t. the test (hold-out set of data) which has been used for training? Can you also add the metric of test vs. train data in all the cases for comparison (Figs. 10., 11)?
> > >
> > > Thanks for pointing this out. We have updated the figure with clearer labels, showing the metric computation with respect to the test data and now including test vs. train comparisons. For Fig. 10 (DDM), the difference between test vs. train is only slightly smaller than the test vs. test baseline. For Fig. 11 (X-rays), there is a noticeable discrepancy between train and test, clearly captured by all metrics. Interestingly, the best-performing model shows a similar distance to test as the train set, indicating it might not fully generalize to the test set (but the results also indicate that there might be a small distribution shift between test and train).
> > >
> > > > Lastly, the community also actively works on developing novel metrics/measures of dissimilarity to compare probability distributions. It may be beneficial for the current survey/guide to at least mention them. For example, paper [3] proposes an alternative to FID based on central kernel alignment (CKA). Paper [4] proposes some topology-based metrics, etc. More broadly, I think it would be beneficial for the current paper to at least mention such and related developments in the field to tell the practitioner some other directions to look at which may be particularly useful for his type of data and problem.
> > >
> > > We thank the reviewer for pointing out this relevant literature (lines 94-104). We added references to other surveys in the introduction. Furthermore, we discuss recently introduced measures of dissimilarity (CKA, Mauve (lines 639-645), topology-based (line 653),  and KID (lines 330-336)).
> > >
> > >
> > >
> > >
> > >
> > >
> > > [1]  Ishan Deshpande, Yuan-Ting Hu, Ruoyu Sun, Ayis Pyrros, Nasir Siddiqui, Sanmi Koyejo, Zhizhen Zhao, David Forsyth, and Alexander G Schwing. Max-sliced Wasserstein distance and its use for GANs. In Proceedings of the IEEE/CVF Conference on Computer Vision and Pattern Recognition, 2019.
> > >
> > > [2]  Jiqing Wu, Zhiwu Huang, Dinesh Acharya, Wen Li, Janine Thoma, Danda Pani Paudel, and Luc Van Gool. Sliced wasserstein generative models. In Proceedings of the IEEE/CVF Conference on Computer Vision and Pattern Recognition (CVPR), June 2019.
> > >
> > > [3] Biwei Dai and Uros Seljak. Sliced iterative normalizing flows. In Proceedings of the 38th International Conference on Machine Learning, volume 139 of Proceedings of Machine Learning Research, pp. 2352–2364. PMLR, 18–24 Jul 202
> > >
> > > [4] Linhart, Julia, et al. "Diffusion posterior sampling for simulation-based inference in tall data settings." arXiv preprint arXiv:2404.07593 (2024).
> > >
> > > [5] Vetter, Julius, et al. "Sourcerer: Sample-based Maximum Entropy Source Distribution Estimation." arXiv preprint arXiv:2402.07808 (2024).
> > >
> > > [6] Deshpande, Ishan, Ziyu Zhang, and Alexander G. Schwing. "Generative modeling using the sliced wasserstein distance." Proceedings of the IEEE conference on computer vision and pattern recognition. 2018.

---

### Review · Reviewer_DPTc · 2024-05-11

**Summary Of Contributions:**

This work experimentally investigated the performance of four measures, Sliced Wasserstein distance, C2ST, MMD, and FID, for measuring the dissimilarity between real distribution and distribution generated by generative models.

**Audience:**

Yes

**Claims And Evidence:**

Yes

**Requested Changes:**

Please see the above weakness section.

**Strengths And Weaknesses:**

## Strength
- This paper is well-written and provides a good summary of existing measures in Section 2.

## Weakness
- The reviewer believes that the results provided by this study do not represent any new findings. This paper only summarized the existing methods and provided their performance for generative models.
- Furthermore, the experiment in Fig. 9 for ImageNet shows the results with only one specific generative model, and Fig. 11 shows the results with only two models. It is unclear whether these results are consistent with the other generative models or specific to these models.
- It is not discussed which distance is a good measure of the distance for the generative model. Since a "good" measure for generative models is one that is consistent with human perception, the "good" measure does not need to distinguish generated images from real images if humans recognize these generated images as real images. However, this work only discussed whether the various measures can distinguish the generated distributions and did not discuss which measure is appropriate.

---

> ### Author Response · Authors · 2024-06-10
> **Rebutal DPTc**
>
> Thank you for your thorough review and constructive feedback. We appreciate your insights and the opportunity to improve our paper. We are happy that the reviewer acknowledges that the “paper is well-written and provides a good summary of existing measures.”
> Below, we will address the weaknesses and requested changes the reviewer pointed out:
>
> > The results provided by this study do not represent any new findings. This paper only summarized the existing methods and provided their performance for generative models.
>
> Indeed, the main goal of our manuscript is not to provide new findings but rather to provide the reader with a guide to understanding commonly used sample-based statistical distances. We believe this exposition has significant value, as it serves as an entry point to a vast and complex literature for researchers and practitioners.
> In light of your comments, we have rewritten the second half of our introduction (lines 85-104) to clarify the scope and intent of our paper. However, we would like to highlight that our original paper included new experiments: we systematically compared MMD, C2TST, FID, and SW across multiple use cases and demonstrated their applications to scientific problems. Additionally, we have now included new experiments comparing more generative models for images and additional kernels for MMD (see next response).
>
> > The experiment in Fig. 9 for ImageNet shows the results with only one specific generative model. Fig. 11 shows the results with only two models. It is unclear whether these results are consistent with the other generative models or specific to these models.
> We thank the reviewer for this valuable comment. We agree that including more generative models for comparison would provide a broader perspective and substantially strengthen our work. Therefore, we have extended our analysis for ImageNet images to include additional generative models.
>
> We generated additional images using another unconditional image generation model [1]. Additionally, we utilized a recently published million-scale dataset of AI-generated images from various generative models [2]. From this dataset, we selected seven additional models: BigGAN [3], ADM [4], GLIDE [5], VQDM [6], Wukong [7], Stable Diffusion [8], and MidJourney [9], each contributing 160,000 images to our evaluation. After embedding all images, we ran our evaluation pipeline using various metrics. As Fig. 9 is intended to demonstrate the specific characteristics of the metric rather than compare different models, we instead summarized the new results in Section 3.3 (lines 457-475) and included the extended analysis in a new appendix section (Appendix A.11), summarizing all results in a table (Table S3) for clarity.
> In the applications section (Figs. 10 and 11), our goal in comparing two models is to illustrate how the presented distances apply to concrete scientific applications, not to derive general insights with respect to these metrics. This provides readers with practical examples of using these metrics for comparing generative models, supported by the provided notebooks.
>
> > It is not discussed which distance is a good measure of the distance for the generative model. Since a "good" measure for generative models is one that is consistent with human perception, the "good" measure does not need to distinguish generated images from real images if humans recognize these generated images as real images. However, this work only discussed whether the various measures can distinguish the generated distributions and did not discuss which measure is appropriate.
>
> The reviewer raises a valid point. Indeed (especially) for images, an important criterion for a good model is whether humans consider its samples real. However, this is not enough, for two main reasons. First, there are use cases where a human potentially will not be able to judge as quickly whether a generated sample looks real or not (e.g., complex time series, protein structures), but we would still like the distribution of generated samples to match the data distributions. Second, just generating realistic samples does not imply we model the whole distribution - if a model just captures one mode well, it will generate realistic-looking samples, but they may lack diversity if the model rarely samples from other modes (See [11], and our Supp Fig 1).
> As we motivated previously (and now clarify in our introduction), a clear-cut answer to which distance to use is impossible given the scope of our article - as we intend this to be a general introduction. Given a specific use case, the question can be answered to some degree. We now more explicitly link to reviews that compare different metrics for a specific use case (lines 94-104).
> The only clear-cut answer we can give based on our experiments (and in line with previous works), is that there is value in using multiple metrics, as a model doing well on one is not a guarantee of it performing well on another metric.

---

> > ### Author Response · Authors · 2024-06-10
> > **Rebutal DPTc part II**
> >
> > [1] Song, Yang, et al. "Consistency models." (2023).
> >
> > [2] Zhu, Mingjian, et al. "Genimage: A million-scale benchmark for detecting ai-generated image." (2024).
> >
> > [3] Brock, Andrew, Jeff Donahue, and Karen Simonyan. "Large scale GAN training for high fidelity natural image synthesis." (2018).
> >
> > [4] Dhariwal, Prafulla, and Alexander Nichol. "Diffusion models beat gans on image synthesis." (2021): 8780-8794.
> >
> > [5] Nichol, Alex, et al. "Glide: Towards photorealistic image generation and editing with text-guided diffusion models." (2021).
> >
> > [6] Rombach, Robin, et al. "High-resolution image synthesis with latent diffusion models. 2022 IEEE." CVF Conference on Computer Vision and Pattern Recognition (CVPR). 2021.
> >
> > [7] Gu, Shuyang, et al. "Vector quantized diffusion model for text-to-image synthesis. 2022 IEEE." CVF Conference on Computer Vision and Pattern Recognition (CVPR). 2021.
> >
> > [8] Wukong, https://xihe.mindspore.cn/modelzoo/wukong. 2022.
> >
> > [9] Rombach, R.; Blattmann, A.; Lorenz, D.; Esser, P.; Ommer, B. High-Resolution Image Synthesis With Latent Diffusion Models. Proceedings of the IEEE/CVF Conference on Computer Vision and Pattern Recognition (CVPR). 2022
> >
> > [10] Midjourney, https://www.midjourney.com/home/. 2022.
> >
> > [11] Theis et al. “A note on the evaluation of generative models” (2016)

---

### Review · Reviewer_oFeX · 2024-05-12

**Summary Of Contributions:**

The paper presents an comprehensive overview of sample-based distance metrics for evaluating generative models.
It covers four types most-commonly used metrics: sliced-Wasserstein distance (SW), classifier-based two-sample test (C2ST), maximum mean discrepancy (MMD) and Frechet Inception distance (FID).
It gives a very good overview (background + related work + intuition) to these metrics and discuss their pros and cons.
It conducts well-designed controllable experiments to study the statistical properties (i.e. how the metrics works when the number of samples and dimensionality change) as well as discusses the corresponding computational cost for practical use.
Finally it studies two real use cases with guidance on how to interpret/use the metrics to analysis models in practice.
I believe this manuscript is a good introduction to evaluation of generative models for people in the ML for science community.

**Audience:**

Yes

**Claims And Evidence:**

Yes

**Requested Changes:**

The main thing missing I think is the relationship between C2ST and density ratio estimation (DRE) and generative adversarial networks (GANs). I suggest the authors to include the following related work in section 2.2
- DRE: the learned classifier has a density ratio interpretation of modeling $r(x) = p(x)/q(x)$ [1] and there are actually more methods than only using neural networks to learn such a classifier or to produce the density ratio estimates (e.g. kernel-mean embedding can be used to produce a method with closed-form solution [1,2])
- Other f-divergences can be estimated based on density ratio estimators as all they needed are an estimated of $r(x)$ and they are used to learn various generative models effectively, e.g. Jensen–Shannon divergence in GANs [3,4], Kullback–Leibler divergence in [5,6], Pearson divergence in [7], etc.
- Especially, the link of the classifier to the discriminator in GANs should be made clear.

[1] Sugiyama, Masashi, Taiji Suzuki, and Takafumi Kanamori. Density ratio estimation in machine learning. Cambridge University Press, 2012.

[2] Huang, Jiayuan, Arthur Gretton, Karsten Borgwardt, Bernhard Schölkopf, and Alex Smola. "Correcting sample selection bias by unlabeled data." Advances in neural information processing systems 19 (2006).

[3] Goodfellow, Ian, Jean Pouget-Abadie, Mehdi Mirza, Bing Xu, David Warde-Farley, Sherjil Ozair, Aaron Courville, and Yoshua Bengio. "Generative adversarial networks." Communications of the ACM 63, no. 11 (2020): 139-144.

[4] Uehara, Masatoshi, Issei Sato, Masahiro Suzuki, Kotaro Nakayama, and Yutaka Matsuo. "Generative adversarial nets from a density ratio estimation perspective." arXiv preprint arXiv:1610.02920 (2016).

[5] Titsias, Michalis K., and Francisco Ruiz. "Unbiased implicit variational inference." In The 22nd International Conference on Artificial Intelligence and Statistics, pp. 167-176. PMLR, 2019.

[6] Huszár, Ferenc. "Variational inference using implicit distributions." arXiv preprint arXiv:1702.08235 (2017).

[7] Srivastava, Akash, Kai Xu, Michael U. Gutmann, and Charles Sutton. "Generative ratio matching networks." arXiv preprint arXiv:1806.00101 (2018).

**Strengths And Weaknesses:**

## Strengths
The draft is well-written, comprehensive and easy to follow.
The introduction and analysis of the metrics are systematic and provides practical guidance on how to use them.
I believe it will be useful for new researchers to apply generative models to real-world, scientific problems in a more systematic way.

## Weaknesses
The focus of the paper is sample-based distances but it was only made clear in the end of the 3rd paragraph in the introduction, i.e. neither the title or the introduction actually makes it clear that's the focus (and therefore these four distances are selected).

---

> ### Author Response · Authors · 2024-06-10
> **Rebutal oFeX**
>
> We thank the reviewer for their kind words and positive assessment of our work, and we appreciate the suggested improvements, which we have incorporated in the revised draft.
>
> > Earlier mention of sample-based distances.
>
> We agree that explicitly stating earlier that our guide is concerned only with sample-based distances provides a more appropriate framing for the paper. We have now mentioned this both in the abstract and earlier in the introduction (lines 85-93), immediately after the overview paragraph on examples of generative models.
>
> > Relationship between C2ST and density ratio estimation and GANs.
>
> Thank you very much for your insightful comment and suggested references. We agree that our review paper should incorporate the relationship among C2ST, DRE, and GANs. We have added a paragraph about the relationship; in particular, we have noted that the learned classifier in C2ST can be used in density ratio estimation and GANs in Section 2.2 of the revised paper (line 242).

---

### Author Response · Authors · 2024-05-24
**Reminder: deadline extension for rebuttal**

Dear Action Editor, dear Reviewers,

Now that the Neurips deadline has past, we are working on revising the paper and addressing all the points you raised — which we are confident will result in a stronger paper.

We would thus like to ask you to please wait with posting your decision till after June 9th.

Best wishes,
The authors

---

### Author Response · Authors · 2024-06-10
**General response**

We would like to thank all reviewers for their constructive engagement with our work, as well as their encouraging comments and suggestions. In particular, we appreciate the positive assessments remarking that our guide was “well-written” (oFeX, DPTc), “comprehensive” (DPTC) / “a good summary” (DPTC), and an overall acknowledgment of the comparison experiments we conducted on toy and real datasets and the recommendations we provided (all reviewers).

In particular, we are happy to hear that our “analysis of the metrics are systematic and provides practical guidance [...]”, and that it “will be useful for new researchers to apply generative models to real-world, scientific problems in a more systematic way” (oFeX). This is exactly the goal we had in mind when writing this introductory and didactic guide on commonly used, sample-based distances in ML for science.

Likewise, we appreciate the constructive criticisms and extensive suggestions from the reviewers. Detailed responses are under individual reviews, and we here provide a summary of the major changes, which we believe have improved the results, framing, and overall presentation of the paper.

**New experiments on additional generative models for images (DPTc)**

We have extended our ImageNet analysis to include additional generative models, generating samples from an additional unconditional model (trained on ImageNet). Furthermore, to evaluate whether the distances are sensitive to differences between ImageNet images vs. other photorealistic images, we utilized a recently published million-scale dataset of AI-generated images from which we selected seven additional models for comparison (including text2image generative models such as Stable Diffusion or Midjourney). These images may be more appealing to human viewers but do not necessarily align with (the entire) distribution of ImageNet images (although they were “prompted” to generate ImageNet classes). We summarize these new results in Section 3.3 (lines 457-476), and include the extended analysis in Appendix A.11 and Table S3.

**Complexity results and discussion of theoretical details (Yh3B)**

Our paper intentionally focuses on the practical side of these metrics rather than diving deep into theory. However, we acknowledge that a discussion of, e.g., sample complexity and bias in sample-based estimators is important and within scope. We have expanded discussions of this (similar to MMD) for the metrics where this property was missing (e.g., lines 147-151, Appendix A.3 & A.4), summarized in Table 1 some key properties, and moved relevant discussions from the Appendix to the main text where bias is discussed for SWD (line 147). Furthermore, we have included information about practical compute time and sample complexity in Appendix A.8 and A.9. Table S2 lists the compute time for each of the scaling experiments presented in Figure 8.

**Additional MMD kernel experiments and discussions (Yh3B)**

We agree that alternative kernels have not been explored sufficiently but might be particularly valuable to practitioners. We performed additional experiments to assess the performance of such kernels, e.g., a linear kernel and the suggested energy-based kernel (the kernel induced by the Euclidean distance), which indeed performs comparatively well and might be easier to apply for practitioners. We have added these results in Appendix A.10, Figure S8, with the kernel comparisons. We additionally mention the link between MMD and energy distance in the main text (lines 285-289). Related, we indeed use Kernel Inception Distances as MMD-based Inception Distance but did not introduce the term properly in the manuscript. We have now added an additional explanation about the relation between KID and FID (Section 2.4, lines 330). Furthermore, as commonly used, we also include MMD with a polynomial kernel in Table S3 (PyTorch Lightnings KID).

---

> ### Author Response · Authors · 2024-06-10
> **General response part II**
>
> **Clarifying discussions of related distances, scope relative to existing surveys/reviews, etc.**
>
> We have added a paragraph properly clarifying the scope of the study (line 85). We have also included further discussions of relationships between specific distances we considered and related concepts, e.g., C2ST in relation to density ratio estimation and discriminator in GANs (oFeX) (Section 2.2, line 242), as well as a brief survey of active works developing metrics of distribution dissimilarity (e.g., CKA, topology-based metrics, Mauve, etc., Section 5, line 531). We have also discussed mode coverage properties in Section 3.5 (line 498).
>
> More generally, in response to reviewer Yh3B’s suggestion, we now explicitly discuss the contribution of our guide relative to existing reviews of statistical distances for generative models in the introduction (line 94). To briefly summarize, existing reviews on distances for generative models are typically more specialized, often considering only a specific class of distances (e.g., divergences) or domains of application or data-type (e.g., images) while being aimed at a more advanced audience. We agree that, in those cases, the distances covered and the corresponding discussions can be more comprehensive. In contrast, however, our guide aims to be an introductory resource that explains and assesses exemplary distances from different classes of distances frequently used in multiple scientific domains that employ generative models.
>
> We do not claim that our survey is better, but that we have a different goal and target a different audience than those of existing surveys. Similarly, we do not aim to provide new findings (DPTc), but to compare commonly used and important tools in generative ML (however, we note that such a systematic and quantitative comparison on the same datasets does not currently exist). We believe such a resource is equally valuable for machine learning practitioners, especially early ones. To be both comprehensive, systematic, and include detailed didactic explanations would require a much bigger text, and we now more explicitly state our narrowed scope in both the introduction and discussion.

---

### Decision · Action_Editor_fYRk · 2024-07-25

**Recommendation:** Accept with minor revision

**Comment:**

The reviewers emphasized that the paper is a survey paper that can serve as an entry point for readers not familiar with the subject. All three reviewers were inclined to accept the paper. One reviewer expressed concern that the new additions during the rebuttal phase somewhat decreased the natural flow of the paper. Acceptance with minor revisions is recommended:

1) Please double-check the spelling of the entire document. For example, see line 587: "comparisons.ue" should be "comparisons due."
2) Please consider improving the natural flow of the paper, which was somewhat disturbed after the rebuttal revision.
3) Please discuss more explicitly the standard limitations of the slicing-based method, when differences in two distributions are concentrated in a small number of features, making a randomly chosen direction in high-dimensional space almost always orthogonal to all of them. Currently, this is only briefly mentioned in Section 5, "Discussions," whereas Section 2.4, "Network-based measures," concerning another method, contains an explicit "Limitations" paragraph.

**Audience:**

yes

**Claims And Evidence:**

yes